# Establishment and Characterization of Free-Floating 3D Macrophage Programming Model in the Presence of Cancer Cell Spheroids

**DOI:** 10.3390/ijms241310763

**Published:** 2023-06-28

**Authors:** Ksenija Korotkaja, Juris Jansons, Karina Spunde, Zhanna Rudevica, Anna Zajakina

**Affiliations:** Cancer Gene Therapy Group, Latvian Biomedical Research and Study Centre, Ratsupites Str. 1, k.1, LV-1067 Riga, Latvia; ksenija.korotkaja@biomed.lu.lv (K.K.);

**Keywords:** macrophage plasticity, tumor-associated macrophages, 3D cell culture models, macrophage reprogramming, cancer cell spheroids, chemokines, cytokines, macrophage polarization

## Abstract

Reprogramming of tumor-associated macrophages (TAMs) is a promising strategy for cancer immunotherapy. Several studies have shown that cancer cells induce/support the formation of immunosuppressive TAMs phenotypes. However, the specific factors that orchestrate this immunosuppressive process are unknown or poorly studied. In vivo studies are expensive, complex, and ethically constrained. Therefore, 3D cell interaction models could become a unique framework for the identification of important TAMs programming factors. In this study, we have established and characterized a new in vitro 3D model for macrophage programming in the presence of cancer cell spheroids. First, it was demonstrated that the profile of cytokines, chemokines, and surface markers of 3D-cultured macrophages did not differ conceptually from monolayer-cultured M1 and M2-programmed macrophages. Second, the possibility of reprogramming macrophages in 3D conditions was investigated. In total, the dynamic changes in 6 surface markers, 11 cytokines, and 22 chemokines were analyzed upon macrophage programming (M1 and M2) and reprogramming (M1→M2 and M2→M1). According to the findings, the reprogramming resulted in a mixed macrophage phenotype that expressed both immunosuppressive and anti-cancer immunostimulatory features. Third, cancer cell spheroids were shown to stimulate the production of immunosuppressive M2 markers as well as pro-tumor cytokines and chemokines. In summary, the newly developed 3D model of cancer cell spheroid/macrophage co-culture under free-floating conditions can be used for studies on macrophage plasticity and for the development of targeted cancer immunotherapy.

## 1. Introduction

Tumor-associated macrophages (TAMs) are a major leukocyte population in the tumor microenvironment (TME). TAMs support tumor progression by inhibiting the anti-tumor immune response, induction of tumor growth, and angiogenesis [1]. Conditionally, macrophages are divided into classically activated macrophages (M1) and alternatively activated macrophages (AAM or M2). High infiltration of the immunostimulatory M1 phenotype is associated with a Th1-dominant anti-tumor immune response and, therefore, a favorable prognosis [2]. M1 can be polarized upon stimulation with interferon-γ (IFNγ) and Toll-like receptor (TLR) agonists such as lipopolysaccharide (LPS) and synthetic lipopeptide Pam3SCK4 [3,4]; this state can be characterized by highly expressed CD38 [5], MHC II, and inducible nitric oxide synthase (iNOS), as well as by the secretion of large amounts of interleukins (IL)-1β, IL-6, IL-12, and tumor necrosis factor-α (TNFα) [5,6,7]. The immunosuppressive M2 phenotype can be induced by the anti-inflammatory cytokines IL-4, IL-10, and IL-13; this state is characterized by highly expressed arginase-1 (Arg1), scavenger receptor CD163, and mannose receptor CD206 [8,9]. A high M2-to-M1 ratio is associated with a poor cancer prognosis.

Macrophage reprogramming toward the M1 phenotype is a promising immunotherapy approach for cancer treatment [10]. Reprogramming can be achieved using pro-inflammatory cytokines, such as TNFα, IL-12, and IFNγ [11,12,13,14]. In previous studies, we reported that the alphavirus-based IFNγ therapy showed a significant decrease in the population of cells bearing myeloid cell markers CD11b, CD38, and CD206 in a murine triple-negative breast cancer model [15]. Even though in vivo therapy reduces the volume of the tumor and increases the M1-to-M2 ratio, the mechanism of macrophage polarization and the factors orchestrating macrophage phenotypes remain unclear.

Traditionally used cell co-cultivation in monolayers does not represent tumor architecture, cell–cell and cell–matrix contacts, or nutrient and oxygen gradients, and therefore is insufficiently appropriate for cancer cell–macrophage interaction studies [16]. Furthermore, passaging/transferring macrophages is practically impossible after differentiation in monolayers, such manipulations can traumatize them and reduce their viability. This problem can be solved using 3D approaches, such as the spheroid model [17]. Spheroid formation using round-bottom ultra-low attachment (ULA) wells, agarose-coated microwells, and the hanging drop technique was reported [18]. The cultivation of macrophages with cancer cell spheroids under free-floating non-adherent conditions may represent a significant advantage over traditional monolayer co-cultures [19]. 

This study aimed to investigate macrophage interactions with cancer cells and to evaluate reprogramming abilities towards the M1 phenotype in vitro under free-floating 3D conditions. For this purpose, we established a murine mammary cancer 3D model with macrophages at different polarization states (non-treated or M0, M1, and M2). The model was characterized by determining the changes in cell surface marker expression and the secretome composition. We hypothesized that cancer cells switch macrophage polarization towards the M2 phenotype. M2 was expected to be involved in cancer cell migration as well as in the direct promotion of tumor growth. Reprogramming toward the M1 phenotype was expected to reverse these processes. This study represents the first attempt to culture macrophages under 3D free-floating conditions with the comprehensive quantitative characterization of surface markers and cytokine/chemokine profiles. This novel approach allows for the examination of cellular behavior and programming properties within a more physiologically relevant environment, deviating from traditional 2D culture systems. The assessment of the macrophage phenotypes co-cultured with cancer cell spheroids potentially will facilitate the modeling of spatial cellular interactions for therapeutic TME programming.

## 2. Results

### 2.1. BMDMs Programming under Free-Floating 3D Conditions

Conventional 2D cultivation of mouse bone marrow-derived macrophages (BMDMs) does not represent in vivo cell–cell interactions and architecture [16,18]. Furthermore, the detachment of 2D (monolayer) polarized macrophage traumatizes cells because standard trypsinization techniques are not applicable to macrophage cultures. In this study, we have investigated the ability of BMDMs to polarize under 3D free-floating conditions. To represent free-floating conditions, the macrophages were detached and plated in a 96-well ultra-low attachment plate (2.9 × 10^4^ cells per well) to achieve an undifferentiated M0 phenotype. A total of 50 ng/mL IFNγ and 100 ng/mL TLR2/1 ligand Pam3SCK4 were added to achieve the M1 phenotype, and 20 ng/mL IL-4 was used to achieve the M2 phenotype. Changes in the cell morphology were detectable after 48 h: M1 macrophages formed net-like structures, while M2 macrophages formed cell clumps, compared to M0 (Figure 1a). When 72 h had passed after programming, the nitric oxide (NO) and secreted cytokines were quantified in cell media, and flow cytometry analysis of cell markers was performed by cell staining with the respective antibodies (Figure 1b–d). 

M1 macrophages were found to express higher levels of CD38 (96%), secrete NO (230 μM), and a high amount of the following pro-inflammatory cytokines: IL-12 (>4.0 ng/mL), TNFα (0.6 ng/mL), and IL-6 (9.8 ng/mL), as described in previous studies of macrophages cultured in monolayers [6,20]. Although M0, M1, and M2 macrophages expressed similar levels of CD11b (Appendix A), M1 polarized macrophages had lower amounts of CD11b^hi^ expressing cells (M1 vs. M2 = 73% vs. 93%, *p* = 0.0039; Figure 1b). The M2 phenotype was identified by an increase in the mannose receptor CD206 (10%) and intracellular arginase-1^hi^ (Arg1^hi^; 50%). MHC II was upregulated in both M1 and M2 types of macrophages with predominant expression in M1 (Figure 1c).

All macrophages were found to secrete IL-1β, IL-2, and IL-16 (Figure 1d and Appendix A). Interestingly, M2 macrophages showed an increased level of IL-1β. Even though IL-1β is conventionally referred to as M1 cytokine [6,21], several studies have shown that M2 cells are also able to secrete it [22,23,24]. These findings suggest that the secretion of IL-1β is not limited to M1 macrophages and that M2 macrophages can also contribute to its production under free-floating conditions. Nevertheless, we suppose that further studies are necessary to assess the IL-1β increase in M2 macrophages under free-floating conditions.

Although both M1 and M2 were able to produce IFNβ, the M1 medium had significantly higher levels of IFNβ (M1 vs. M2 = 2.1 vs. 1.0 ng/mL, *p* = 0.0070). Previous studies showed that IFNβ has a growth inhibitory effect on cancer cells [25], upregulates the percentage of CD11b^low^ macrophages, promotes the expression of IL-10, and reduces the expression of IL-12 [26]. Also, both M1 and M2 produced similar amounts of GM-CSF compared to undifferentiated M0 macrophages (Appendix A). No or very low levels of IFNα and IL-10 were detected similar to previous findings (Appendix A) [3,27].

M2 programming provoked macrophage proliferation. M2 macrophages had significantly higher cell numbers compared to M0 (Appendix A, *p* = 0.0254, Day 4), as IL-4 is a well-known proliferative agent [28]. In contrast, M1 had significantly lower viability and reduced cell number compared to M0 (*p* < 0.0001), as the M1 activating factor IFNγ has an antiproliferative effect (Appendix A) [29]. Furthermore, NO induces oxidative stress. In summary, the obtained data show the successful programming of BMDMs under free-floating conditions.

### 2.2. BMDMs Reprogramming under Free-Floating 3D Conditions

The reprogramming of M2 macrophages towards the M1 phenotype is a promising cancer immunotherapy method. To perform reprogramming in a 3D free-floating model, BMDMs were activated for 48 h as described above, and then the NO test of cell media was performed to control M1 polarization (Figure 2a,b, Day 2). Next, the cells were washed and an equal number of cells was transferred to a medium without activation factors to eliminate the possibility that both programming and reprogramming factors are affecting cells at the same time. After 24 h, the reprogramming factors were added (Figure 2a M1→M2 and M2→M1). A total of 50 ng/mL IFNγ and 100 ng/mL Pam3SCK4 were used to achieve M2→M1 phenotype, and 20 ng/mL IL-4 was used to achieve M1→M2. Groups containing M1 or M2 macrophages cultivated in the medium without activating factors (*w*/*o*) and not reprogrammed were stated as reprogramming-negative controls (Figure 2a (*w*/*o*) M1 and (*w*/*o*) M2). The reprogramming continued for 72 h, and on Day 6 of the experiment, the cells were visualized by microscopy, and the surface markers and secretome were analyzed using flow cytometry, ELISA, or Luminex, respectively. 

Reprogramming led to morphological changes in 3D cultured macrophages (Figure 2b): M1→M2 macrophages increased in size and stopped forming M1-like nets, whereas M2→M1 formed a net and clumps at once. (*w*/*o*) M1 and (*w*/*o*) M2 were spread evenly in the wells. The cultivation of M1 without IFNγ/Pam3SCK4 ((*w*/*o*) M1) decreased the NO levels significantly (from 150 μM to 37 μM, *p* ≤ 0.0001; Figure 2b Nitric oxide). Interestingly, (*w*/*o*) M2 showed a decrease in the CD206 marker, indicating possible depolarization in the absence of an IL-4 stimulus.

Reprogramming was achieved as M2→M1 started to produce NO (330 μM) and express less CD11b^hi^ compared to (*w*/*o*) M2 (69% vs. 97%, *p* = 0.0003; Figure 2c). Furthermore, M2→M1 started secreting pro-inflammatory cytokines IL-12 (>4 ng/mL), TNFα (1.2 ng/mL), and IL-6 (8 ng/mL), achieving an M1-like phenotype (Figure 2d). Although both reprogrammed phenotypes expressed MHC II and CD38 (Figure 2c), M2→M1 had higher levels of MHC II^hi^ (M1→M2 vs. M2→M1 = 17% vs. 52%, *p* = 0.0002) and CD38^hi^ (M1→M2 vs. M2→M1 = 14% vs. 28%, *p* < 0.0001) (Appendix A). Furthermore, M1→M2 started to express M2 marker CD206 (M1→M2 vs. (*w*/*o*) M1 = 13% vs. 0.2%, *p* = 0.0086) (Figure 2c). 

Remarkably, both reprogrammed phenotypes expressed high intracellular levels of arginase 1 (Arg1^hi^), representing a characteristic feature of M2 macrophages; the proportions of non-reprogrammed (*w*/*o*) M1 and (*w*/*o*) M2 Arg1^hi^-positive cells were 4% and 5%, respectively, while reprogrammed M1→M2 and M2→M1 were 27% and 33% (*p* = 0.0251 and *p* = 0.0720). Furthermore, the reprogrammed phenotypes began to secrete higher amounts of IFNβ (Figure 2d): 0.3 ng/mL was detected in the medium of non-reprogrammed (*w*/*o*) M1, and 1.9 ng/mL in the medium of reprogrammed M1→M2 macrophages (*p* = 0.0062). Similarly, in non-reprogrammed (*w*/*o*) M2 samples, the IFNβ was undetectable (<62.5 pg/mL), but for M2→M1, it was 1.36 ng/mL (*p* < 0.0001). Moreover, both reprogrammed phenotypes secreted higher amounts of GM-CSF (Appendix A): 0.34 pg/mL and 0.28 pg/mL were detected in the medium of non-reprogrammed (*w*/*o*) M1 and (*w*/*o*) M2, while in the medium of reprogrammed M1→M2 and M2→M1, the amounts were 0.6 pg/mL and 0.8 pg/mL (*p* = 0.0091 and 0.0026). IFNα and IL-10 were not detected (Appendix A). 

The induction of pro-inflammatory cytokines and metabolic markers (Arg1) may be the result of cellular overstimulation during reprogramming. On the other hand, as macrophages possess high plasticity, we suppose that they might lose phenotype features when the activation factors are removed [30,31]. Thus, (*w*/*o*) control macrophages had lower levels of secreted cytokines, except for IL-1β. Similar findings were previously described by Tarique et al. [32].

Although reprogramming under free-floating conditions is possible, both M1→M2 and M2→M1 macrophages achieved mixed phenotypes expressing MHC II, Arg1^hi^, CD38, secreting IFNβ, IL-1β, IL-2, and IL-16 (Figure 2c,d and Appendix A). These findings show that macrophages do not change their phenotype completely following reprogramming, and some features from the previous phenotype are still present. The profiles of all those mentioned above and tested macrophage phenotype markers, cytokines, and chemokines are summarized in Section 2.6, Figure 7. The results of the chemokine analysis are described in Section 2.5: “Chemokines secreted by macrophages and 4T1/GFP spheroids”.

### 2.3. BMDMs Programming in the Presence of Cancer Cell Spheroids

It is widely accepted that cancer cells influence/stimulate tumor macrophages to acquire pro-tumoral phenotype. However, the mechanism of such “education” is unclear. Here, we tested the changes in macrophages induced by cancer cells and evaluated their potential programming abilities in the presence of cancer cell spheroids under free-floating conditions. First, the 4T1/GFP breast cancer cell spheroids were generated, and then the non-treated M0 macrophages were added (Figure 3a). BMDMs were incubated with cancer cell spheroids for 24 h before programming to achieve a potential TAM-like phenotype. Then, the programming factors (M1 and M2) were added, as described above. Later, 72 h after programming, morphological changes were recorded; M0 and M2 macrophages induced the migration of cancer cells out of the spheroid, and cancer cell satellite spheroids were formed (Figure 3b). 

To distinguish the cytokines produced by cancer cells, controls without macrophages were performed. The control spheroids were incubated without activation factors (S), with M1 activation factors IFNγ/Pam3SCK4 (S IFNγ Pam3), and with M2 activation factor IL-4 (S IL-4). We evaluated the secretion of IL-1β, IL-2, IL-6, IL-10, IL-12, IL-16, GM-CSF, IFNβ, and TNFα (Appendix A). It was found that upon stimulation with IFNγ/Pam3SCK4, the spheroids secreted pro-inflammatory cytokines IL-16 (102 pg/mL), IL-6 (0.25 ng/mL), and GM-CSF (50 pg/mL); upon stimulation with IL-4, the spheroids secreted IL-2 (S vs. S IL-4 = 1.0 vs. 1.9 pg/mL, *p* = 0.0020) and GM-CSF (30 pg/mL). Furthermore, low levels of GM-CSF (7 pg/mL) were detectable even in a non-stimulated spheroid medium (S), which was previously described in the 4T1 cell line [33]. It was concluded that the cancer cell spheroid secretome is affected by IFNγ/Pam3SCK4 and IL-4. 

The analysis of macrophages co-cultured with 4T1/GFP spheroids (S + M0; S + M1; S + M2) demonstrated that macrophages can be programmed by respective IFNγ/Pam3SCK4 and IL-4 stimuli (Figure 3c,d and Appendix A). After 72 h programming, S + M1 macrophages started to express MHC II (70%), CD38 (92%), fewer CD11b^hi^ (44%), and secreted NO (270 μM), TNFα (1.5 ng/mL), IL-6 (10.2 ng/mL), and IL-12 (>4.0 ng/mL). S + M2 macrophages were marked by the expression of CD206 (10%) and Arg1^hi^ (50%). Similar to the macrophage 3D monoculture, both S + M1 and S + M 2 secreted IFNβ (S + M1: 1.8 ng/mL; S + M2: 3.0 ng/mL) and IL-1β (S + M0: 130 pg/mL; S + M1: 107 pg/mL; S + M2: 180 pg/mL) (Figure 3d), but no IFNα or very low IL-10 were detected in co-culture (Appendix A). The obtained extracellular and intracellular marker profiles, as well as the cytokine profile, were similar to those shown in Figure 1c,d (programming without cancer cell spheroids). It was concluded that macrophages can be programmed in the presence of cancer cell spheroids.

### 2.4. BMDMs Reprogramming in the Presence of Cancer Cell Spheroids

Here, we investigated the potential macrophage reprogramming abilities in the presence of cancer cell spheroids. Initially, macrophages were programmed without 4T1 spheroids in a 3D monoculture to acquire M1 and M2 phenotypes. Then, the macrophages were washed and transferred to the cancer cell spheroids, incubated for 24 h, and the reprogramming factors were added (Figure 4a). Briefly, IFNγ/Pam3SCK4 was used to obtain an S + M2→M1 phenotype, and IL-4 was used to obtain the S + M1→M2 phenotype. M1 and M2 macrophages, to which no reprogramming factors were added, are stated as macrophages S + (*w*/*o*) M1 and S + (*w*/*o*) M2, respectively, and used as controls.

After 72 h of reprogramming in the presence of 4T1/GFP spheroids, flow cytometry analysis was performed, and the amount of secreted cytokines and chemokines was determined. S + M1→M2 began to express the M2 marker CD206 (22%), while S + M2→M1 began to express the M1 marker CD38 (76%), fewer CD11b^hi^ (40%; Figure 4c), and produce NO (230 μM; Figure 4b), TNFα (0.7 ng/mL), IL-6 (9.3 ng/mL), and IL-12 (>4 ng/mL; Figure 4d). Therefore, it was concluded that macrophages were reprogrammed in the presence of cancer cell spheroids. Similar to reprogramming without cancer cell spheroids, macrophages began to produce Arg1^hi^ (S + M1→M2: 34%; S + M2→M1: 37%) and IFNβ (S + M1→M2: 2.6 ng/mL; S + M2→M1: 2 ng/mL) (Figure 4c,d), but IFNα and IL-10 cytokines were not detected (Appendix A). In addition, both phenotypes expressed MHC II (S + M1→M2: 50%; S + M2→M1: 80%), CD38 (S + M1→M2: 57%; S + M2→M1: 76%), and IL-1β (S + M1→M2: 99 pg/mL; S + M2→M1: 120 pg/mL) (Figure 4c,d). Interestingly, S + M1→M2 induced the formation of satellite cancer cell spheroids, whereas S + M2→M1 did not stimulate cancer cell migration similar to the M1 phenotype (Figure 3b and Figure 4b). 

### 2.5. Chemokines Secreted by Macrophages and 4T1/GFP Spheroids

Chemokines are important for immune cell recruitment and immune control. In this study, macrophage reprogramming resulted in new, previously undescribed macrophage phenotypes, M1→M2 and M2→M1. Furthermore, we observed cancer cell (4T1/GFP) migration out of the spheroids in the presence of M0 and M2, but not M1 macrophages. To explain the cancer cell migration and formation of spheroid satellites, as well as to further characterize the reprogrammed macrophage phenotypes, we assessed the secreted chemokine profiles under 3D free-floating conditions. First, the chemokines secreted by cancer cell spheroids without macrophages and activation factors were determined (Figure 5a). The 4T1/GFP spheroids (S) secreted CCL5 (53 pg/mL) and CXCL5 (290 pg/mL), which are associated with a negative cancer prognosis [34,35]. The spheroids also secreted CXCL16 (181 pg/mL), which was also described in previous studies and is considered a positive prognosis [36]. 

Chemokines secreted by the cancer cell spheroids without macrophages under the influence of activating factors were then determined. Upon stimulation with IFNγ/Pam3SCK4 (S IFNγ Pam3), the spheroids started to secrete chemokines induced by IFNγ—CCL5 (0.6 ng/mL), CXCL10 (3.4 ng/mL), and CXCL11 (190 pg/mL) (Figure 5a). Increased expression of CXCL10 after treatment of human breast cancer cells with IFNγ was previously described by Fluhr et al. [37]. Compared to unstimulated spheroids (S), after stimulation with IFNγ/Pam3SCK4, the spheroid began to produce CCL3 (38 pg), CCL20 (45 pg/mL), CCL22 (1.4 pg/mL), CXCL1 (3.4 ng/mL), CXCL5 (5 ng/mL), and CX3CL1 (150 pg/mL) (Figure 5a). 

After stimulation with IL-4, the spheroids began to secrete CCL5 (0.17 ng/mL), CCL17 (20 pg/mL), CCL11 (5 pg/mL), and CCL19 (7 pg/mL) (Figure 5a). Importantly, BMDMs medium supplemented with 10% L929-CM also contained the monocyte chemotactic factors CCL2 (2.6 ng/mL), CCL7 (1.6 ng/mL), and VEGF-A (2.4 ng/mL), neutrophil chemotactic factor CXCL1 (0.7 ng/mL), and eosinophil chemotactic factor CCL11 (2.6 pg/mL) (Appendix A). 

Then, the chemokines secreted by M1 macrophages in a free-floating monoculture (M1) and co-culture with cancer cell spheroids (S + M1) were determined (Figure 5b). M1 revealed high amounts of IFNγ-induced chemokines: CCL5 (2.3 ng/mL), CXCL10 (22 ng/mL), and CXCL16 (716 pg/mL) (Figure 5b). M1 also secreted the monocyte chemotactic factor VEGF-A (4.9 ng/mL), which is in agreement with previous studies [38]. M1 macrophages expressed the neutrophil chemotactic factors CXCL1 (1 ng/mL), CXCL5 (210 pg/mL), CXCL11 (3.8 ng/mL), CCL12 (3 ng/mL), and CCL24 (8 ng/mL). Interestingly, M1 secreted a significant amount of the M2 macrophage-specific chemokine CCL22 (1.1 ng/mL), which was previously observed in M1-like MHC II^hi^ TAMs in vivo [39]. In addition, the secretion of CCL12, CCL22, and CCL24 was higher in the group without cancer cell spheroids (M1), indicating an inhibitory effect of spheroids on these chemokines. The effect of the spheroids on the amounts of chemokines secreted by macrophages is described in detail in the Section titled “The influence of cancer cell spheroids on BMDMs”. The levels of the other tested chemokines are illustrated in Appendix A.

Continuing the characterization of programmed macrophages, the chemokines secreted by M2 macrophages in 3D monoculture (M2) and co-culture with cancer cell spheroids (S + M2) were determined (Figure 5b). M2 secreted an increased amount of CCL1 (2.7 pg/mL) as well as the following IL-4-induced chemokines: CCL17 (0.3 ng/mL), CCL22 (1.5 ng/mL), and CCL24 (21 ng/mL). CCL17 and CCL22 attract type II T helpers (Th2) and regulatory T cells (Tregs); thus, these cytokines are associated with immunosuppressive TME and poor prognosis [40]. Macrophage production of CCL22 was also observed in human ovarian cancer [41]. Contrary to the observations in the M1 macrophage group, the CCL12 level was high only in the presence of cancer cell spheroids (M2: 0.8 ng/mL; S + M2: 3.4 ng/mL, *p* = 0.0004). CCL12 was also upregulated in M2-like MHC II^low^ TAMs [39] and in the presence of 4T1-CM [33], suggesting that the established 3D co-culture model reliably represents the properties of TAMs.

Although the macrophages lost several phenotypic characteristics after the elimination of activation factors (*w*/*o* M), the expression of some chemokines was remarkably stable (Figure 6). Pre-activated M1 macrophages incubated without activation factors (*w*/*o*) continued to secrete CCL5 (2.3 ng/mL), CXCL1 (0.6 ng/mL), CXCL5 (80 pg/mL), CXCL16 (0.8 ng/mL), and VEGF-A (2.7 ng/mL). The levels of the other tested chemokines are illustrated in Appendix A.

Reprogramming of M1 macrophages into an M2 phenotype resulted in M1→M2, which similarly to (*w*/*o*) M1 continued to secrete the following M1-specific chemokines: CCL5 (2.2 ng/mL), CXCL1 (0.34 ng/mL), CXCL16 (0.63 ng/mL), VEGF-A (0.2 ng/mL), and CCL1 (1.9 pg/mL). After M1→M2 reprogramming, the M2-associated chemokine CCL22 (0.72 ng/mL) began to be secreted. Unlike M2 macrophages, M1→M2 began to secrete certain IL-4-induced chemokines only after co-culture with cancer cell spheroids, for instance, CCL17 (M1→M2: 0.02 ng/mL; S + M1→M2: 0.12 ng/mL, *p* = 0.0004) and CCL24 (M1→M2: 0.5 ng/mL; S + M1→M2: 2.4 ng/mL, *p* = 0.0016) (Figure 6).

The expression of M2-associated chemokines CCL17 (24 pg/mL), CCL22 (0.6 ng/mL), and CCL24 (2.7 ng/mL) remained stable in M2 macrophages incubated without factors (*w*/*o* M2) (Figure 6). Also, after M2→M1 reprogramming, M2 cytokines CCL17 (45 pg/mL), CCL22 (1.3 ng/mL), and CCL24 (3.3 ng/mL) continued to be secreted (Figure 6). Because the reprogrammed phenotype continues to secrete chemokines that attract immature dendritic cells (iDC) and Th2, namely, CCL17 and CCL22 [42], reprogramming as a therapeutic strategy may fail. However, analysis of the secreted chemokines of the reprogrammed phenotype revealed that M2→M1 after reprogramming began to secrete M1-associated factors CXCL1 (1.2 ng/mL), CXCL5 (200 pg/mL), CXCL10 (9 ng/mL), CXCL11 (1.1 ng/mL), CXCL16 (0.7 ng/mL), CCL5 (2.4 ng/mL), CCL12 (1 ng/mL), and VEGF-A (1.4 ng/mL) (Figure 6).

### 2.6. The Influence of Cancer Cell Spheroids on BMDMs

This study represents a comprehensive evaluation of cytokine, chemokine, and surface marker profiles of macrophages co-cultured with cancer cell spheroids, as well as an assessment of their reprogramming potential. In this Section, we have highlighted the data demonstrating the influence of 4T1/GFP cancer cell spheroids on macrophage surface/intracellular markers and secretome profiles. Figure 7a summarizes the obtained surface marker and secretome characteristics of programmed/reprogrammed macrophages in the presence/absence of cancer cell spheroids. The heatmap data visualization presented in Figure 7b allows us to clearly identify the cell signatures that are stably induced by programming and the features that emerge transiently only in the presence of activation factors and disappear without them ((*w*/*o*) M1, (*w*/*o*) M2: without factors IFNγ/Pam3SCK4 and IL-4, respectively). For example, CD38 and CCL5 are stably expressed by M1 macrophages, including (*w*/*o*) M1 and M1→M2, and are slightly affected by the presence of cancer cell spheroids. In contrast, IL-12 and CXCL10 in M1 are highly dependent on the presence of IFNγ/Pam3SCK4 in the medium and are completely downregulated in (*w*/*o*) M1. 

**Markers.** How do 4T1/GFP spheroids affect the BMDMs phenotype? After incubation of M0 with cancer cell spheroids (S + M0) for 96 h (Figure 3a, schematic representation), the macrophages demonstrated elevated levels of CD38 (from 5% (M0) to 17% (S + M0), *p* = 0.0038; Figure 8a, Non-polarized). Although CD38 is a good marker of M1 macrophages in vitro [7], several studies have associated the presence of CD38 in TME with a negative prognosis [43]. Furthermore, in the macrophage-cancer cell co-culture, the levels of macrophage MHC II were downregulated compared to M0 cultivated alone (from 23% to 4.5%, *p* = 0.0003, Figure 8b). The decrease in MHC II levels after BMDMs cultivation with 4T1 conditioned media was previously reported by Madera et al. [44]. 

The macrophages that were cultivated with cancer cell spheroids for 24 h before polarization and then polarized in the presence of spheroids (S + M1; S + M2) had decreased levels of CD11b^hi^ (M1: from 73% to 44%, *p* = 0.0022; M2: from 93% to 56%, *p* = 0.0106) and MHC II (M1: from 91% to 70%, *p* = 0.0069; M2: from 70% to 34%, *p* = 0.0242) (Figure 8c). In addition, S + M1 had increased levels of Arg^hi^ from 0.1% to 2.1%, *p* = 0.0057 (Figure 8a, Programming).

Macrophages reprogrammed in the presence of cancer cell spheroids had decreased levels of CD11b^hi^ (M1→M2: from 94% to 67%, *p* < 0.0001; M2→M1: from 69% to 40%, *p* = 0.0116). S + M1→M2 had downregulated levels of MHC II from 68% to 50%, *p* = 0.0007, and CD38 from 75% to 57%, *p* = 0.0007, but upregulated levels of CD206 from 13% to 22%, *p* = 0.0734 (Figure 8a, Reprogramming). We suppose that co-cultivation with cancer cell spheroids reduces the ability of macrophages to induce an immune response via antigen presentation due to a decrease in MHC II levels. Moreover, a decrease in the proportion of cells expressing the integrin CD11b^hi^ can negatively affect the phagocytic activity of macrophages.

**Cytokines.** After incubation of M0 with cancer cell spheroids for 96 h, the macrophages demonstrated elevated levels of IFNβ (from 42.4 to 376 pg/mL, *p* = 0.0767), while the levels of NO were downregulated compared to M0 cultivated in the 3D monoculture (from 13 to 9 μM, *p* = 0.0081) (Figure 8d, Non-polarized). The levels of TNFα, IL-1β, IL-6, and IL-12 were not affected by the cancer cell spheroids (Appendix A).

S + M1 had increased levels of TNFα (from M1 0.6 ng/mL to S + M1 1.5 ng/mL, *p* = 0.0206) (Figure 8d Programming). Previous studies showed that macrophages cultivated with 4T1-CM secreted significantly larger amounts of TNFα after stimulation with LPS [44]. Macrophages cultured with the estrogen receptor-positive (ER^+^) cancer cell line T47D also expressed more TNFα [45]. Interestingly, M2 in co-culture secreted decreased level of IL-12 (from 171 to 90 pg/mL, *p* = 0.0026; Appendix A) and a larger amount of IFNβ than in the 3D monoculture (from M2 1 ng/mL to S + M2 3 ng/mL, *p* = 0.0021) (Figure 8d, Programming). 

M1→M2, which were reprogrammed in the presence of spheroid demonstrated increased IFNβ levels (from 1.9 to 2.6 ng/mL, *p* = 0.0234). M2→M1 macrophages had statistically significant downregulation in NO expression (from 330 to 230 μM, *p* = 0.0121) and TNFα (from 1.2 to 0.7 ng/mL, *p* = 0.0179; Figure 8d, Reprogramming). M2 macrophages cultured without activation factors (*w*/*o*) after interaction with cancer cell spheroids had increased IL-1β (from (*w*/*o*) M2 90 pg/mL to S + (*w*/*o*) M2 136 pg/mL, *p* = 0.0009), IFNβ (from 0 to 0.6 ng/mL, *p* = 0.0300), and TNFα (from 98 to 131 pg/mL, *p* = 0.0007) cytokine secretion (Appendix A).

**Chemokines.** Since several chemotactic factors, such as CXCL5, CXCL16, and VEGF-A, were also detected in the cancer cell spheroid 3D monoculture (Figure 5a and Appendix A), their changes were difficult to evaluate. After 96 h of M0 incubation with cancer cell spheroids, CXCL1 was increased in their culture medium (from M0 0.21 ng/mL to S + M0 0.31 ng/mL, *p* = 0.0187) (Figure 5b). Similar observations have been previously described in co-culture with MDA-MB-231 and T47D [45]; in addition, CXCL1 secreted by TAMs promotes metastasis [46]. On the other hand, the changes between M0 and S + M0 were detected for chemokines CXCL12 (from 0.09 to 0.06 ng/mL, *p* = 0.0102), CXCL16 (from 0.58 to 0.45 ng/mL, *p* = 0.0012), and CCL22 (from 0.37 to 0.11 ng/mL, *p* = 0.0037), demonstrating the inhibitory effect of the cancer cell spheroids (Figure 5b).

M1 macrophages co-cultured with cancer cell spheroids showed increased levels of CXCL1 (from M1 1 ng/mL to S + M1 3 ng/mL, *p* = 0.0377), CCL20 (from 5 to 50 pg/mL, *p* = 0.0096), and CX3CL1 (from 60 to 136 pg/mL, *p* = 0.0015, Appendix A), which was likely due to the interaction of 4T1/GFP cancer cells with the M1 polarization factors IFNγ/Pam3SCK4 (Figure 5a,b). In the co-culture S + M1, the following chemokines were decreased: CCL4 (from M1 8.0 ng/mL to S + M1 2.6 ng/mL, *p* = 0.0007), CCL12 (from 3 to 1.7 ng/mL, *p* = 0.0641), CCL22 (from 1.1 to 0.4 ng/mL, *p* = 0.0501), CCL24 (from 8 to 2 ng/mL, *p* = 0.0093), and CXCL11 (from 3.8 to 2.7 ng/mL, *p* = 0.0226) (Appendix A and Figure 5b). Previously, a decrease in CXCL11 gene expression was observed in macrophages co-cultured with ovarian cancer cells [47]. It is unclear whether the reduction is due to the binding of chemokines to cancer cell receptors or due to the inhibition of chemokine production. 4T1 is known to express the CCL17 and CCL22 receptors CCR4 [48], as well as the CXCL12 receptors CXCR4 and CXCR7 [49]. Thus, the reduction in CXCL12 and CCL22 in the presence of cancer cell spheroids may be partially explained by the binding of these chemokines to their receptors.

M2 macrophages polarized in the presence of spheroids (S + M2) had increased levels of chemokines CXCL1 (from M2 0.14 ng/mL to S + M2 0.45 ng/mL, *p* = 0.0013) and CCL12 (from 0.8 to 3.4 ng/mL, *p* = 0.0004). Increased levels of CX3CL1 (from 75 to 115 pg/mL, *p* = 0.0056) and CCL11 (from 13 to 20 pg/mL, *p* = 0.0077) were also observed in S + M2 (Appendix A), which can be explained by the interaction of 4T1/GFP cancer cells with the M2 activation factor IL-4 (Figure 5a).

M1 macrophages cultured without activation factors (*w*/*o*) after interaction with the cancer cell spheroid showed increased CCL20 (from (*w*/*o*) M1 3 to S + (*w*/*o*) M1 15 pg/mL, *p* = 0.0013) and CXCL1 (from 0.6 to 1.5 ng/mL, *p* = 0.0032) levels, decreased CCL3 (from 1.5 to 0.2 ng/mL, *p* = 0.0205) and CXCL13 (from 0.20 to 0.11 ng/mL, *p* = 0.0144) (Appendix A). A decrease in CCL3 gene expression was also detected in macrophages after co-culture with ovarian cancer cells [47]. On the other hand, in colorectal cancer, CCL20 secreted by TAMs is associated with the recruitment of Tregs [50]. S + (*w*/*o*) M2 macrophages showed increased levels of CXCL1 (from (*w*/*o*) M2 0.13 ng/mL to S + (*w*/*o*) M2 0.32 ng/mL, *p* = 0.0175) and CCL24 (from 2.7 to 9 ng/mL, *p* = 0.0016). It is theoretically possible that a leaky amount of activation factors (IL4, or IFNγ/Pam3SCK4) are present in the (*w*/*o*) macrophage environment and may interact with the cancer cell spheroid. However, taking into account the loss of the macrophage phenotypic characteristics, we suppose that the levels of the remaining activation factors are too small to affect chemokine production.

Analysis of the reprogrammed phenotypes revealed that M1→M2 macrophages in co-culture with spheroids had increased amounts of CXCL1 (from M1→M2 0.34 ng/mL to S + M1→M2 1.36 ng/mL, *p* = 0.0002), CCL17 (from 0.02 to 0.12 ng/mL, *p* = 0.0004), CCL20 (from 2.7 to 11.7 pg/mL, *p* = 0.0005), CCL22 (from 0.72 to 0.92 ng/mL, *p* = 0.0026), CCL24 (from 0.5 to 2.4 ng/mL, *p* = 0.0016), and CX3CL1 (from 65 to 120 pg/mL, *p* = 0.0060) (Figure 6 and Appendix A). The increase in CX3CL1 chemokine levels can be attributed to the interaction of 4T1/GFP cancer cells with IL-4 (Figure 5a). Increased macrophage CCL22 gene expression has been shown previously after co-culture with ovarian cancer cells [47], and Curiel et al. showed that CCL22 secreted by TAMs recruits Tregs in human ovarian carcinoma and promotes tumor development [41].

In contrast, M2→M1 macrophages reprogrammed in the presence of cancer cell spheroids had increased levels of CCL17 (from M2→M1 45 pg/mL to S + M2→M1 113 pg/mL, *p* = 0.0021) and CCL24 (from 3.3 to 5 ng/mL, *p* = 0.0356) (Figure 6). The increase in chemokines such as CCL20 (from 4 to 33 pg/mL, *p* = 0.0010), CX3CL1 (from 64 to 143 pg/mL, *p* = 0.0001), CXCL1 (from 1.2 to 3.6 ng/mL, *p* = 0.0010), and CXCL10 (from 9 to 18 ng/mL, *p* = 0.0061) in the S + M2→M1 culture medium can be attributed to the interaction of 4T1/GFP cancer cells with IFNγ/Pam3SCK4 because an increase in these chemokines was observed in spheroids cultured in the presence of IFNγ/Pam3SCK4 without macrophages (Figure 5a).

Changes in cell surface markers and secretome profiles after the co-culture of macrophages with cancer cell spheroids are summarized in Table 1. Overall, CXCL1 was elevated in all samples of cancer cell spheroids. However, it is unclear whether the CXCL1 is elevated due to macrophage–spheroid interactions or due to the spheroid response to IFNγ/Pam3SCK4. Yoshimura et al. described that GM-CSF induces macrophage CXCL1 and CCL17 production [33]. Since spheroids secrete a small amount of GM-CSF in the absence of activation factors and higher amounts in the presence of activation factors (Appendix A), they can induce macrophage expression of CXCL1 and CCL17.

### 2.7. BMDMs Do Not Affect the Spheroid Size

The size of the cancer cell spheroids was measured using fluorometry and by counting total cell fluorescence in microscopy images. Relative spheroid sizes were determined using flow cytometry (total cell number). The results of all three methods were similar, supporting the same tendency (Figure 9a).

Spheroids cultivated with M1 and M2→M1 macrophages were significantly smaller than the spheroids cultivated with M0 macrophages (Figure 9a, Fluorometry, *p* = 0.0026 and *p* < 0.0001, respectively). As IFNγ has a direct antiproliferative effect on cancer cells [51], we used controls that included spheroids with IFNγ/Pam3SCK4 or IL-4 but without macrophages. The spheroids cultivated with IFNγ/Pam3SCK4 were significantly smaller than those without the same factors (Figure 9a, Fluorometry, *p* < 0.0001). Furthermore, the spheroids cultivated with IFNγ/Pam3SCK4 had lower viability than the spheroids cultivated in the absence of factors (*p* = 0.0136) and spheroids cultivated with IL-4 (*p* = 0.0151), which can be explained by the toxic effect of these factors (Figure 9b).

The effect of M1 and M2→M1 macrophages on spheroid size in the presence of IFNγ/Pam3SCK4 was not significantly different from the spheroids with IFNγ/Pam3SCK4 factors without macrophages (*p* > 0.05). Although macrophages were functionally programmed (or reprogrammed) to express pro-inflammatory markers and cytokines, they were not capable of decreasing the size of the cancer cell spheroids in this model.

### 2.8. 4T1 Cells and Macrophages Migrate toward Each Other

To study cell migration, 8.0 µm pore inserts were used. The migration of cancer cells to macrophages in different polarization states was determined. The cancer cells were able to migrate to M0 and M2, and migrated significantly less to M1 plated in a standard monolayer culture 24-well plate (M2 vs. M1 *p* = 0.0171; Figure 9c). Similarly, the formation of satellite cancer cell spheroids was observed in the co-culture with M0 and M2 (Figure 3b and Figure 4b). In addition, M2 macrophages secreted CCL17 and CCL22 (Figure 5b) ligands for the receptor CCR4 expressed by 4T1 cells [48]. A macrophage culture medium was used as a negative control in the migration assay. The cancer cells did not migrate to the culture medium, suggesting that cancer cell migration was induced only in the presence of macrophages. Interestingly, in the literature, 4T1/GFP cells migrated to the M1 macrophage culture medium [52].

Next, we examined the ability of M0, M1, and M2 macrophages to migrate toward 4T1/GFP cancer cells. For this, cancer cells were seeded in a 24-well plate, while BMDMs were added to the inserts with M0, M1, or M2 activation medium. Macrophages were able to migrate to cancer cells through an 8.0 µm pore membrane, but not to the 4T1/GFP culture medium (Figure 9d). M1 and M2 migrated more significantly than M0 (*p* < 0.001). Xuan et al. showed that CXCL10 induces chemotaxis in M1 macrophages [53]. As we found after stimulation of cancer cells with IFNγ/Pam3SCK4, cancer cells begin to secrete CXCL10 (Figure 5a). In addition, Vogel et al. demonstrated that M2 macrophages migrate to CCL5 [27], which is secreted at low levels by cancer cells after stimulation with IL-4 (Figure 5a).

## 3. Discussion

In this study, we have developed and characterized a macrophage programming and reprogramming model in the presence of cancer cell spheroids under 3D free-floating conditions. In total, 6 surface/intracellular markers, 11 cytokines, and 22 chemokines were analyzed, providing a comprehensive characterization of the phenotypic and functional properties of BMDMs. Many studies on macrophage profiling are focused on the specific markers, cytokines, or chemokines [24,54,55,56,57,58]. The advantage of this study is that it represents the global assessment of macrophage programming features that are monitored in terms of the presence/absence of the respective stimuli. Moreover, for the first time, the reprogramming of BMDMs co-cultured with cancer cell spheroids under free-floating conditions was characterized and the respective changes were determined. 

A few studies have shown that macrophages can be reprogrammed in vitro [32,59], and some studies have described reprogramming in vivo [10,58,60,61]. Although the principal possibilities of macrophage reprogramming have been partially investigated, no studies that explain the reprogramming process in detail have been conducted. The majority of conclusions are based on the presence of a few markers and functional features. It is insufficient to draw conclusions regarding the cells with high plasticity. In this study, the phenotypes of M1→M2 and M2→M1 reprogrammed macrophages were investigated by analyzing the surface markers, amounts of secreted cytokines and chemokines. It was concluded that reprogramming leads to a mixed phenotype expressing both M1 and M2 markers. M1→M2 continued to express the M1 marker CD38 and the respective chemokines CXCL1, CXCL16, CCL5, and CCL12 (Figure 4c and Figure 6). On the other hand, M2→M1 continued to express the M2 characteristic marker Arg1^hi^ and the chemokines CCL17 and CCL24 (Figure 4c and Figure 6). Practically, this signature and other features should be considered in the development of reprogramming-based therapy. 

The tumor microenvironment represents a complex and dynamic system where the factors produced by the cancer cells generate an immunosuppressive “cold” state of the immune cells [62,63]. Many aspects of cancer-related immune suppression remain unclear. We used a 3D free-floating model to investigate how the murine mammary cancer cell spheroids affect the programming of macrophages. Madera et al. demonstrated that macrophage exposure to 4T1-secreted factors enhanced macrophage responses to bacteria-derived TLR agonists [44]. As a result, macrophages had higher levels of nitric oxide and pro-inflammatory cytokines, as well as enhanced phagocytosis activity. Our studies showed that co-culture of M1 macrophages with 4T1 cells indeed enhanced the secretion of pro-inflammatory factors such as TNFα after stimulation with the TLR2/1 ligand Pam3SCK4 and IFNγ (Figure 8d). At the same time, our data showed that co-cultivation led to the production of metastasis-promoting chemokine CXCL1 in M1 macrophages (Figure 5b) and downregulated the macrophage (M0, M1, and M2) expression of CD11b^hi^ and MHC II (Figure 8a), the surface molecules that are involved in phagocytosis and antigen presentation [64]. Furthermore, non-polarized M0 macrophages showed increased expression of CD38 after co-culture with cancer cell spheroids (Figure 8a), which can be linked to the myeloid-derived suppressor cell (MDSC)-like phenotypic changes [65,66]. Co-culture of M1 macrophages with cancer cell spheroids significantly increased the expression of the M2-associated marker, Arg1^hi^ (Figure 8a). Previous studies also showed that macrophages co-cultured with non-small cell lung cancer (NSCLC), ovarian cancer, or triple-negative breast cancer (TNBC) cells obtain an immunosuppressive M2-like phenotype [45,47,67]. In summary, our data indicate significant changes in the macrophage phenotype after cultivation with cancer cell spheroids. 

Chemokines are chemotactic cytokines that provide signals for cell migration, which is critical for immune cell composition in tumors [68]. Macrophages can be recruited to the TME by CCL2, CCL3, CCL4, CCL5, CCL7, CCL8, and CXCL12 chemokines [69]. Our findings show that 4T1 cell spheroids secrete chemokines CCL5, CXCL5, CXCL16 (Figure 5a), and GM-CSF (Appendix A). Elevated levels of the chemokine CCL5 and its receptor CCR5 have been found in more than 58% of TNBC and HER-2^+^ subtype breast cancer clinical tumor samples [70]. In addition, it was previously shown that GM-CSF secreted by 4T1 cells induces macrophage CCL2 production [33]. CCL2, known as monocyte chemoattractant protein 1 (MCP-1), attracts monocytes and was shown to be associated with the spread of breast cancer in the body [71]. In this study, we do not see the enhancement of CCL2 production by macrophages co-cultured with cancer cell spheroids, probably because the macrophage cultivation medium supplemented with L929-CM already contained a significant amount of CCL2 (2.6 ng/mL) (Appendix A). 

Cancer cells cultured with macrophage activation factors (IFNγ/Pam3SCK4, or IL-4) change their chemokine profile. Upon stimulation with IFNγ/Pam3SCK4, the spheroids secrete increased levels of IL-6, IL-16, CCL5, CXCL10, and CXCL11 (Appendix A and Figure 5a); and also produce CCL3, CCL20, CCL22, CXCL1, CXCL5, and CX3CL1 (Figure 5a). On the other hand, upon stimulation with IL-4, the spheroids secrete low but significant levels of CCL5, CCL11, CCL17, and CCL19 (Figure 5a). Interestingly, CCL11, known as the eosinophil chemotactic factor, is much higher induced by IL-4 than by IFNγ/Pam3SCK4. The high CCL11 level was shown to increase the proportion of immunosuppressive CD4^+^CD25^+^Foxp3^+^ Tregs in a breast cancer model [72], indicating the pro-tumoral effect of the IL-4/CCL11 interaction. In another study, the expression of IFNγ-induced chemokine CXCL10 in the tumor negatively correlated with tumor size and microvessel density (MVD) but positively correlated with the number of infiltrating CD8^+^ cells [73]. CXCL10 elicits strong inflammatory effects and is also known to recruit the tumor antigen-specific CD8^+^ T cells into tumors; however, its stability and downstream functions in the tumor environment are not clear [74]. Therefore, IFNγ/CXCL10 can be linked to the anti-tumor immune response. Other IFNγ-induced chemokines CCL5, CCL20, and CXCL1 were shown to be related to cancer progression and poor prognosis [34,75,76]. The association of inflammatory chemokines with different prognoses is confusing. The chemokines represent a complex network where one group of chemokines may contribute to acute inflammation, while another one to chronic inflammation [77]. The tumor microenvironment is characterized by chronic inflammation, which does not induce an anti-tumor immune response, but in contrast, it promotes disease progression [78]. The coordination of chemokines and their specific role in cancer cell escape from immune surveillance remain to be elucidated. 

The developed 3D free-floating cell system represents a useful tool for the analysis and modeling of the cytokine/chemokine profiles of immune cells in the presence/absence of cancer cells. Figure 7a,b provides a comprehensive summary of the surface marker and secretome profiles of macrophages that were programmed and reprogrammed in the presence or absence of cancer cell spheroids. These profiles offer valuable insights into the phenotypic and functional properties of macrophages under different conditions. Potentially, the model allows generating more complex heterotypic cultures with macrophages, T-cells, NK-cells, fibroblasts, etc., resembling the native tumor composition. Simultaneous coordination of different cytokine and chemokine levels is critical for the therapeutic programming of immune cells. 

In the established model, macrophages stimulated the migration of 4T1/GFP cancer cells out of the spheroid. In the M2 group, the formation of satellite GFP-positive spheroids was observed. Cancer cells were found to migrate to M0 and M2 macrophages, but not to M1, whereas M2 and M1 macrophages migrated to cancer cells (Figure 9c,d). Thus, the reciprocal migration of M2 macrophages and cancer cells possibly results in satellite spheroid formation. Vogel et al. [27] showed that M2 macrophages migrate to CCL5, which, as we have found, is released by cancer cells following IL-4 stimulation. Therefore, we suppose that the satellite spheroids are formed in a CCL5-dependent manner. It can be concluded that the developed model represents a new free-floating system for cell migration studies where the potential cancer cell distribution can be studied.

Although macrophages were functionally programmed/reprogrammed and expressed inflammatory markers and cytokines, they did not significantly affect the size of the cancer cell spheroids. Furthermore, IFNγ in combination with the TLR2/1 ligand Pam3SCK4 significantly reduced the size of the spheroid without the presence of macrophages. On the other hand, when these factors are removed, macrophages start to lose their phenotypic features. This makes potential therapeutic reprogramming challenging.

Although the established model represents TME better than conventional 2D cultivation, it has certain limitations. First, the volume of the wells is limited and the model does not supply nutrients. As a result, the developed model is not suitable for long-term cell cultivation. The use of microfluidic devices can potentially solve the problem; however, it can be difficult to accomplish under free-floating conditions. Second, the traditional macrophage cultivation protocol requires the use of a medium supplemented with L929-CM, in which monocyte and eosinophil chemotactic factors CCL2, CCL7, VEGF-A, CCL11, and CXCL1 have been found (Appendix A). In this context, the application of synthetic media can be useful. Third, BMDMs represent monocyte-derived macrophages and not tissue-resident macrophages. The molecular profiles of BMDMs and tissue-resident macrophages, as well as TAMs, can differ in response to co-culture with cancer cells [79]. Nevertheless, TAMs in mammary tumors are mostly blood-derived and primarily originate from the recruitment and differentiation of inflammatory CCR2^+^Ly6C^hi^CX3CR1^low^ monocytes [39]. As TAMs are mainly monocyte-derived, BMDMs are suitable for representing the interactions between cancer cells and macrophages. Finally, it is important to mention that the polarized macrophage population is heterogeneous, which means that non-polarized M0 can still be present, as shown by the percentage of cells expressing specific cell surface markers (Figure 1c). Furthermore, the use of the IFNγ co-stimulatory TLR ligand Pam3SCK4 instead of the more standard LPS also needs to be carefully considered. LPS has a non-specific effect on the immune system, thereby potentially leading to unfavorable consequences upon its administration [4,80,81]. On the other hand, Horuluoglu et al. showed that Pam3SCK4 can stimulate monocytes to obtain an M2-like phenotype [82]. The specific effects of LPS, Pam3SCK4, and other TLR ligands should be further evaluated in the proposed model in terms of macrophage phenotyping. 

In summary, a new spheroid-based cancer cell model for macrophage programming under 3D conditions was established and characterized. Unlike matrix-based 3D models, the use of an ultra-low attachment plate allows for the easy manipulation of cells and secreted factors. This model can be used for in vitro immunotherapy studies to develop a tumor micromodel with other cancer-associated cell types, such as fibroblasts or immune cells. The co-culture allowed us to evaluate how cancer cell spheroids affect macrophages in different polarization states, and how macrophages affect spheroid growth. The described model can be used to study TME cell–cell interactions and can be potentially applied as a screening tool for TAM-targeted therapies. 

## 4. Materials and Methods

### 4.1. Cell Lines

Murine triple-negative breast cancer 4T1-Fluc-Neo/eGFP-Puro cells (4T1/GFP) expressing green fluorescent protein (GFP) and firefly luciferase (Luc) were obtained from Imanis Life Sciences. The 4T1/GFP cells were cultivated in Roswell Park Memorial Institute (RPMI) in 1640 medium (Cat. No. 12-115F; Lonza™ BioWhittaker™, Cologne, Germany) supplemented with 10% fetal bovine serum (FBS; Cat. No. FBS-HI-12A; Capricorn Scientific, Ebsdorfergrund, Hessen, Germany), 2 mM L-glutamine (Cat. No. 25030-024; Gibco, Life Technologies, Thermo Fisher Scientific, Waltham, MA, USA), 1% penicillin/streptomycin (PEST) (Cat. No. 15070-063; Gibco, Life Technologies, Grand Island, NY, USA), 0.1 mg/mL G418 (Cat. No. 10131-027; Gibco, Life Technologies), 2 µg/mL puromycin (Cat. No. ant-pr; InvivoGen, San Diego, CA, USA) at 37 °C, and 5% CO_2_ in a humidified atmosphere. Fibroblast-like L929 cells were obtained from ATCC^®^ (CCL-1™) and cultivated in RPMI-1640 medium supplemented with 10% FBS, 1% PEST, 2 mM L-glutamine at 37 °C, and 5% CO_2_ in a humidified atmosphere.

Cells were incubated in a humidified 5% CO_2_ incubator at 37 °C and passaged using 0.05% trypsin solution (Cat. No. 15400-054, Gibco, Life Technologies).

### 4.2. Isolation and Generation of Bone Marrow-Derived Macrophages (BMDMs)

Mouse macrophages were differentiated from bone marrow cells according to an established protocol using an L929 cell-conditioned medium as a source of macrophage colony-stimulating factor (MCSF) [3]. The L929 cell-conditioned medium was prepared as follows: L929 cells were plated in RPMI-1640 with 10% FBS, 2 mM L-glutamine, and 1% PEST in T175 flasks (Cat. No. 156502; Nunc, Thermo Fisher Scientific, Waltham, MA, USA) to achieve 60–70% confluence, the medium was replaced with fresh 30 mL medium, and the cells were incubated for 8–9 days in a humidified 5% CO_2_ incubator at 37 °C. After cultivation, the conditioned medium (CM) of L929 cells was collected, centrifuged at 400× *g* for 10 min, transferred to a new tube, and centrifuged again at 13,000× *g* for 20 min at 4 °C to remove the cell pellets. The CM of L929 cells was aliquoted and stored at −20 °C until use.

Bone marrow from the femurs and tibia of the hind legs of 8–12 weeks old female BALB/c mice was flushed with RPMI-1640 medium containing 10% FBS under sterile conditions and passed through a 70-µm cell strainer (Cat. No. 800070; Bioswisstec, Schaffhausen, Switzerland). Red blood cells (RBC) were removed by incubation in an ammonium–chloride–potassium (ACK) lysing buffer (Cat. No. A10492-01; Gibco, Life Technologies) for 10 min at room temperature. After centrifugation and washing, the remaining cells were cultured in 10 cm non-tissue culture-treated dishes (Cat. No. 0030702018; Eppendorf, Hamburg, Germany) in RPMI-1640 medium (10% FBS) containing 30% L929-cell-CM. Bone marrow cells (Passage 0) were cultured for 5 days, after which non-adherent cells were washed off using phosphate-buffered saline (PBS), and the adherent macrophages were cultured for an additional day. Differentiated BMDMs were detached by incubation in PBS without CaCl_2_ and MgCl_2_ for 30 min at 4 °C and by repeated gentle flushing of the dish, and then the BMDMs were collected by centrifugation. BMDMs were then frozen in Recovery™ Cell Culture Freezing Medium (Cat. No. 12648010; Gibco, Life Technologies, Thermo Fisher Scientific, Waltham, MA, USA) and stored at −150 °C for future experiments.

After thawing, BMDMs (Passage 1) were incubated for 4 days, detached, and filtered through a 70 μM cell strainer. For experiments under 3D conditions, 3.5 × 10^4^ cells were plated in a 96-well black round bottom ultra-low attachment plate (Cat. No. CLS4515, Corning, Life Sciences, New York, NY, USA) in 200 μL of the macrophage cultivation medium: RPMI-1640 medium containing 10% FBS, 50 U/mL penicillin, 50 µg/mL streptomycin, 2 mM L-glutamine, and 10% L929-CM. These cells (Passage 2) were used for polarization and reprogramming experiments. 

### 4.3. BMDMs Polarization into M0/M1/M2 Phenotype

The non-polarized or M0 phenotype was achieved by cultivating cells in RPMI-1640 medium supplemented with 10% FBS (Cat. No. ES-009-B; Millipore, Burlington, MA, USA), 10% MCSF (L929-CM), 1% PEST, and 1% L-glutamine for 72 h. The media was supplemented either with 50 ng/mL recombinant mouse IFNγ (Cat. No. BMS326; eBioscience, San Diego, CA, USA) and 100 ng/mL Pam3SCK4 (Cat. No. tlrl-pms; InvivoGen, San Diego, CA, USA) to achieve M1 phenotype, or with 20 ng/mL IL-4 (Cat. No. RP-8666; Invitrogen, Waltham, MA, USA) to achieve M2 phenotype.

### 4.4. Reprogramming of BMDMs

BMDMs were polarized to the M0/M1/M2 phenotype for 48 h, as described above. Then, cells were collected, washed with PBS, counted, and plated 2.87 × 10^4^ cells in 100 μL per well in a fresh macrophage medium. After 24 h, 50 μL of macrophage cultivation medium supplemented with 50 ng/mL recombinant mouse IFNγ and 100 ng/mL Pam3SCK4 was added to the M2 pre-polarized macrophages and 20 ng/mL IL-4 was added to the pre-polarized M1. The samples (*w*/*o*) M1 and (*w*/*o*) M2 received 50 μL of macrophage cultivation medium to keep the total volumes equal in all wells. 

### 4.5. Formation of Spheroids

4T1-Fluc-Neo/eGFP-Puro (4T1/GFP) spheroids were generated using a 96-well black round bottom ultra-low attachment plate (Cat. No. CLS4515; Corning, Life Sciences, Berlin, Germany). Cells were collected from monolayers by trypsin treatment, filtered through a 40 μM cell strainer (Cat. No. CLS431750; Corning, Life Sciences), counted, and resuspended in RPMI-1640 medium supplemented with 10% FBS, 1% PEST, and 1% L-glutamine. The suspension was added to a 96-well ultra-low attachment plate (3 × 10^3^ cells in 100 μL per well) and incubated at 37 °C in a humidified 5% CO_2_ incubator.

### 4.6. Co-Cultivation of Macrophages with the Cancer Cell Spheroid

When 48 h passed after spheroid formation, M0, M1, and M2 macrophages (polarized as described above for 48 h) were washed, counted, and 2.87 × 10^4^ cells in 100 μL per well were added to the cancer cell spheroid. Macrophages were then co-cultivated with 4T1/GFP spheroids for 24 h before programming or reprogramming was performed. A total of 50 ng/mL recombinant mouse IFNγ and 100 ng/mL Pam3SCK4 were added to achieve M1 and M2→M1 phenotypes, and 20 ng/mL IL-4 was added to achieve M2 and M1→M2 phenotypes.

### 4.7. Flow Cytometry

For flow cytometry, cells from 4 wells were combined, the suspension was centrifuged at 410× *g* for 5 min and washed with PBS. Spheroids were treated with 0.05% trypsin solution for 5 min. Then, the cells were washed with PBS supplemented with 10% FBS (PBS-FS10%) and centrifuged at 410× *g* for 10 min. 

Cells were then washed with PBS and stained with Fixable Viability Stain 660 (FVS 660; Cat. No. 564405, BD Biosciences, Franklin Lakes, NJ, USA) to determine cell viability. Briefly, cells were suspended in FVS 660 in PBS (1: 10,000 according to the manufacturer’s protocol), incubated for 15 min at room temperature (RT), and then cells were washed twice with PBS.

Then, 50 μL of PBS-FS10% supplemented with 12.5 μg/mL mouse IgG (Cat. No. I8765-5MG; Sigma-Aldrich, Merck, Darmstadt, Germany) was added to the cell pellet and incubated for 30 min at 4 °C. After blocking, the cells were washed with 800 μL PBS-FS10% and the suspension was centrifuged at 410× *g* for 10 min. Then, the cell pellet was resuspended in 50 μL PBS-FS10% with fluorophore-labeled monoclonal antibodies diluted as recommended by the manufacturers. The following monoclonal antibodies were used: anti-CD11b-APC-eFluor™ 780 (Cat. No. 47-0112-82; Invitrogen); anti-MHC II-PE (Cat. No. 12-5321-82; Invitrogen); anti-CD206-Briliant Violet 421 (Cat. No. 141717; Biolegend, San Diego, CA, USA); anti-CD38-PerCP-eFluor™ 710 (Cat. No. 46-0381-82; Invitrogen). Cells were incubated for 1 h at 4 °C, after which 800 μL of PBS-FS10% was added, and the suspension was centrifuged for 10 min at 410× *g*. The cell pellet was washed once with PBS-FS10%.

For intracellular staining, PerFix-nc Kit (Cat. No. B31168; Beckman Coulter, Brea, CA, USA) was used. Briefly, cells were resuspended in 25 μL FBS, and 15 μL of the fixative reagent was added. The suspension was mixed and incubated for 15 min at RT, and then 150 μL of permeabilization reagent and anti-Arginase1-APC (Cat. No 17-3697-82; Invitrogen) antibodies were added in appropriate amounts. The suspension was incubated in the dark at RT 40 min. Then, 1.2–1.8 mL of the final reagent (diluted in H_2_O) was added to the suspension, and it was centrifuged for 10 min at 410× *g*. The cell pellet was resuspended in 200 μL of PBS-FS10%. The experiment was repeated twice, and each staining was performed in duplicate. The data were obtained by a FACSAria BD Hardware flow cytometer using FACSDiva Software (BD Biosciences, San Jose, CA, USA). To create the compensation matrix, UltraComp eBeads™ (Cat. No. 01-2222; Invitrogen, Thermo Fisher Scientific) was used. The data were analyzed using FlowJo v10.0.7 software and presented as the mean of two independent experiments.

### 4.8. Analysis of Cytokines and Chemokines by ELISA and Luminex Assays

Supernatants of macrophages programmed and reprogrammed in the presence and absence of cancer cell spheroids were subjected to cytokine and chemokine analyses using ELISA and Luminex assays. The media was combined from at least 4 wells, centrifuged for 5 min at 500× *g*, aliquoted, and stored at −20 °C until use. Then, the cytokines were determined using Mouse IL-12 ELISA Kit (Cat. No. BMS616; Invitrogen), Mouse IFNα ELISA Kit (Cat. No. BMS6027; Invitrogen), Mouse IFNβ ELISA Kit (Cat. No. CSB-E04945m; CUSABIO, Houston, TX, USA), and Mouse VEGF-A ELISA Kit (Cat. No. BMS619-2; Invitrogen). 

For the determination of chemokines, Mouse Chemokine Panel 31-Plex (Cat. No. 12009159; Bio-Rad, Hercules, CA, USA) was used. Briefly, the beads were washed twice, and then 50 μL of the samples and standards were added to the beads. To ensure proper mixing, the plate was covered with foil and incubated at RT on a shaker at 850 rpm for 30 min. Following the incubation, the plate was washed three times before the addition of the detection antibodies. The plate was incubated on a shaker in the dark. After 30 min, the plate was washed three times. Then, streptavidin-phycoerythrin (SA-PE) was added to the plate. The plate was incubated in the dark at 850 rpm for 10 min. After the incubation, the plate was washed three times to remove any excess SA-PE. The beads were resuspended in an assay buffer and shaken for 30 s prior to analysis with Luminex 100/200™ and xPONENT 3.1.971.0 software package for the data acquisition system (Luminex Corporation, Austin, TX, USA).

### 4.9. Nitric Oxide (NO) Assay

A nitric oxide assay kit was used (Cat. No. EMSNO; Invitrogen) to determine the level of NO in the cell culture medium. NO is measured by the determination of nitrites. Briefly, 50 µL of the cell culture medium was collected from each well, centrifuged, and used for NO quantification. The standards provided in the kit were utilized to generate a standard curve and quantify the nitrites. A spectrophotometer was employed to measure optical density at 540 nm.

### 4.10. Migration Assay

The 8.0 µm pore polycarbonate membrane inserts (Cat. No. 3422; Corning, Life Sciences) were used to investigate the migration of cancer cells and the differently activated macrophages. To investigate the migration of cancer cells towards macrophages, BMDMs were plated on the bottom of the 24-well plate (Cat. No. 30024; SPL Life Sciences, Pocheon-si, Gyeonggi-do, Republic of Korea). When the macrophages reached confluency, they were programmed towards the M0/M1/M2 phenotype, as described above. After 4 h, 4 × 10^4^ 4T1/GFP cells in 300 μL complete medium were added to the upper inserts. Then, the inserts were placed into wells containing the differently activated macrophages. 

To investigate the migration of BMDMs towards cancer cells, 5 × 10^4^ 4T1/GFP cells were plated on the bottom of the 24-well plate. After 24 h, the BMDMs were stained with CellTracker™ CM-DiI Dye (Cat. No. C7001; Invitrogen), and 300 μL of 7.5 × 10^4^ cells were added into the upper inserts. The inserts were then inserted into the wells. 

After 18 h, the non-migrated cells were removed using a cotton swab. Migrated cells localized on the opposite membrane site were counted using a Leica DM-IL fluorescent microscope. Cell migration against media (4T1/GFP, M0, M1, and M2 media, respectively) was used as a negative control.

### 4.11. Determination of Cancer Cell Spheroid Growth 

**Fluorometry.** The spheroid growth was measured by a GFP fluorometry assay using Victor3V 1420-040 Multilabel HTS Counter (PerkinElmer): emission filter 485 nm, detection filter 535 nm. 

**Microscopy.** Fluorescent microscopy images were obtained using a Leica DM-IL microscope. Total fluorescence was calculated by ImageJ using the following formula: Total Fluorescence = Area Integrated Intensity − (Area × Background Average Fluorescence) (1)

**Flow cytometry.** The number of GFP-positive cells per sample was determined by a FACSAria II BD Hardware flow cytometer using FACSDiva 6.1.3 Software.

### 4.12. Statistical Analysis

Statistical analysis was performed using the GraphPad Prism 8.02 software. Data were expressed as mean ± standard error of the mean (SEM). Statistical analysis was performed using the Student’s *t*-test and one-way ANOVA. An unpaired one-tailed Student’s *t*-test was used to compare groups. Each experimental group in this study consisted of two biological replicates. To increase the statistical power, each biological replicate was created by pooling four independently treated samples, which were processed under the same experimental conditions. *p* < 0.05 was considered statistically significant (* *p* < 0.05; ** *p* < 0.01; *** *p* < 0.001; ns—non-significant). Statistical analysis of larger groups (*n* ≥ 3) was performed using one-way ANOVA. 

Heatmaps were generated using a normalization approach depending on the type of data being analyzed. For macrophage markers, the scale was based on the percentage of maximal expression, ranging from 0% to 100%. In the case of cytokines and chemokines, a normalized scale ranges from 0 to 1 and represents the relative amount of secreted cytokines.

## 5. Conclusions

In this study, we created a 3D macrophage programming model in the presence of 4T1 breast cancer cell spheroids. The model was characterized using surface markers, secreted cytokines, and chemokines. Unlike many previous studies that focused on specific markers or cytokines, this research offers a global assessment of macrophage signatures.

Macrophage reprogramming was performed, and it was established that reprogramming causes a heterogeneous phenotype that expresses both M1 and M2 markers. In a 3D spheroid model, cancer cells were demonstrated to alter macrophage characteristics, promoting M2-like phenotype features. This novel cancer cell spheroid-based model for macrophage programming in 3D settings can be exploited for in vitro immunotherapy studies. It offers opportunities for more complex heterotypic cultures resembling the native tumor composition and facilitates studies on immune cell programming in a clinically relevant context. 

Although the established model represents a step forward in mimicking the tumor microenvironment, it has certain limitations. Addressing these limitations, such as nutrient supply and the use of synthetic media, will further enhance the relevance and applicability of the model. 

## Figures and Tables

**Figure 1 ijms-24-10763-f001:**
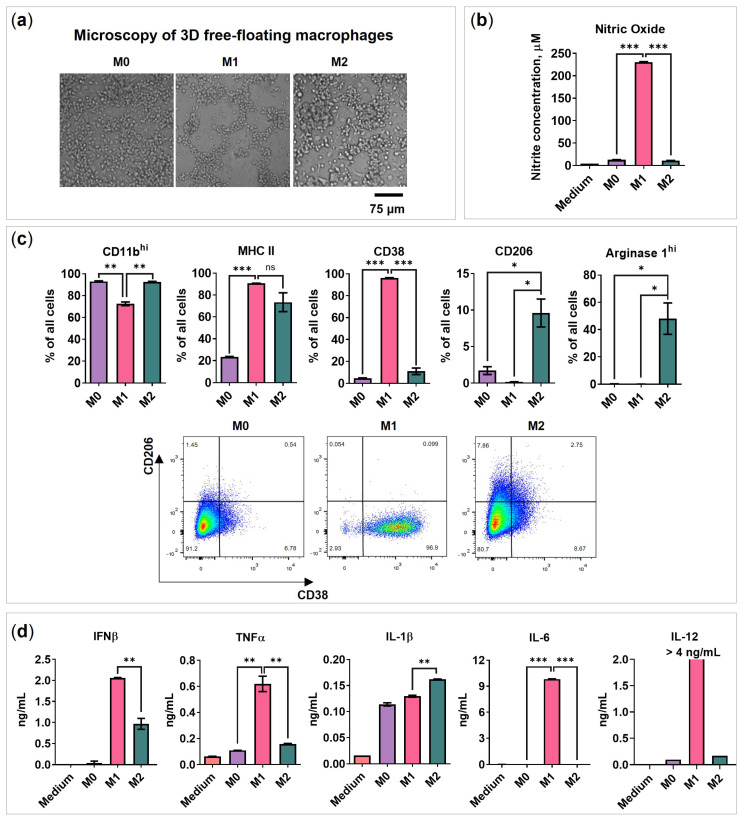
BMDMs programming under free-floating 3D conditions. BMDMs were plated in a 96-well black round bottom ultra-low attachment plate in a medium containing the following programming factors: 50 ng/mL recombinant mouse IFNγ and 100 ng/mL Pam3SCK4 to achieve M1 phenotype or with 20 ng/mL IL-4 to achieve M2 phenotype. M0 represents the initial BMDM phenotype. (**a**) Bright-field microscopy of M0/M1/M2 48 h after activation. (**b**) Nitric oxide test 72 h after activation (*n* = 3). (**c**) Flow cytometry determined marker expression of 3D polarized macrophages 72 h after activation; data are presented as % of all cells. Flow cytometry analysis showing representative staining of CD38 on the *x*-axis and CD206 on the *y*-axis. (**d**) The concentration of cytokines in the medium of 3D polarized macrophages 72 h after activation was determined using ELISA or Luminex assay. Medium—RPMI-1640 medium supplemented with 10% FBS, 50 U/mL penicillin, 50 µg/mL streptomycin, 2 mM L-glutamine, and 10% L929-conditioned medium (CM). Data are shown as the mean of two experiments ± SEM (*n* = 2, each experiment is a pool of 4 biological replicates). * *p* < 0.05; ** *p* < 0.01; *** *p* < 0.001; ns—non-significant.

**Figure 2 ijms-24-10763-f002:**
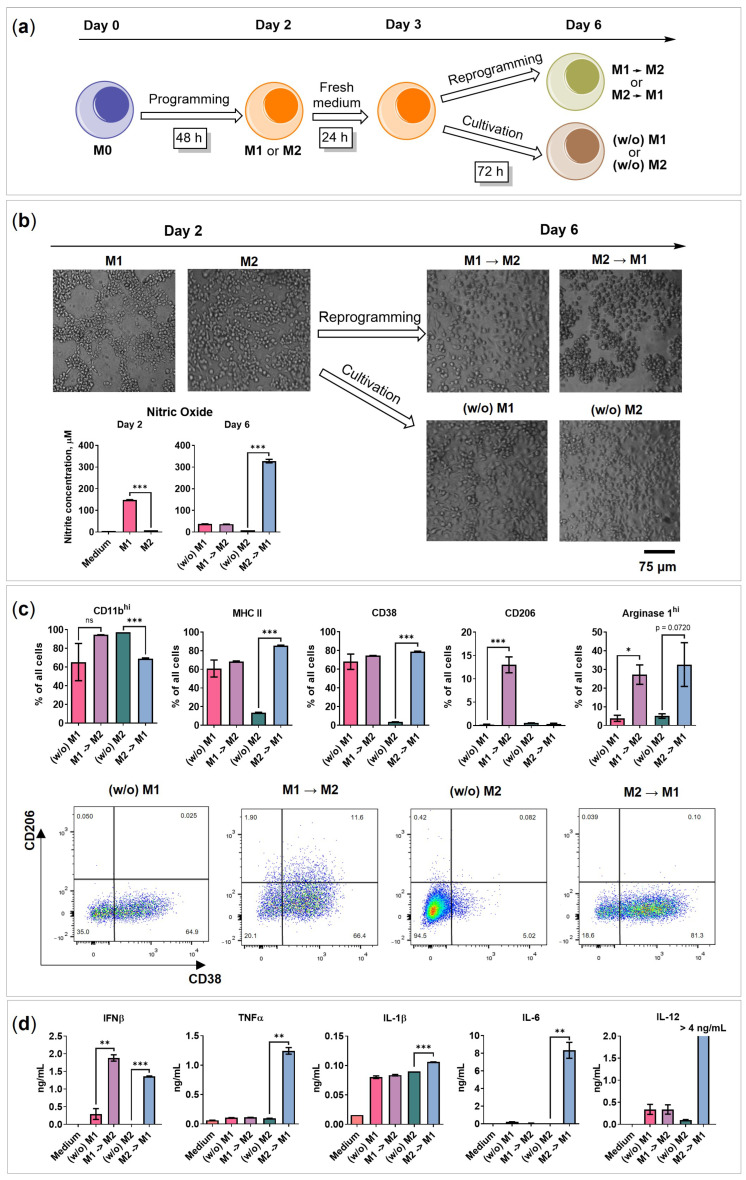
BMDMs reprogramming under 3D free-floating conditions. (**a**) Scheme of reprogramming. BMDMs were polarized into M1 or M2 phenotype for 48 h, then the cells were collected, washed with PBS, and plated in a fresh macrophage medium. On Day 3, 50 ng/mL of recombinant mouse IFNγ and 100 ng/mL Pam3SCK4 were added to the M2 polarized macrophages and 20 ng/mL IL-4 to M1. For achieving (*w*/*o*) M1 and (*w*/*o*) M2 phenotype, cells were cultivated in a medium without factors from Day 2 till Day 6. (**b**) Bright-field microscopy and nitric oxide test of reprogrammed macrophages (M1→M2 and M2→M1) and cultivated macrophages ((*w*/*o*) M1 and (*w*/*o*) M2)) before (Day 2) and after reprogramming/cultivation (Day 6). (**c**) Marker expression of reprogrammed macrophages on Day 6. Cells were analyzed using flow cytometry; data are presented as % of total cells. Flow cytometry analysis showing representative staining of CD38 on the *x*-axis and CD206 on the *y*-axis. (**d**) The concentration of cytokines in the medium of reprogrammed macrophages on Day 6. Medium—RPMI-1640 medium containing 10% FBS, 50 U/mL penicillin, 50 µg/mL streptomycin, 2 mM L-glutamine, and 10% L929-CM. Data are shown as the mean of two experiments ± SEM (*n* = 2, each experiment is a pool of 4 biological replicates). * *p* < 0.05; ** *p* < 0.01; *** *p* < 0.001; ns—non-significant.

**Figure 3 ijms-24-10763-f003:**
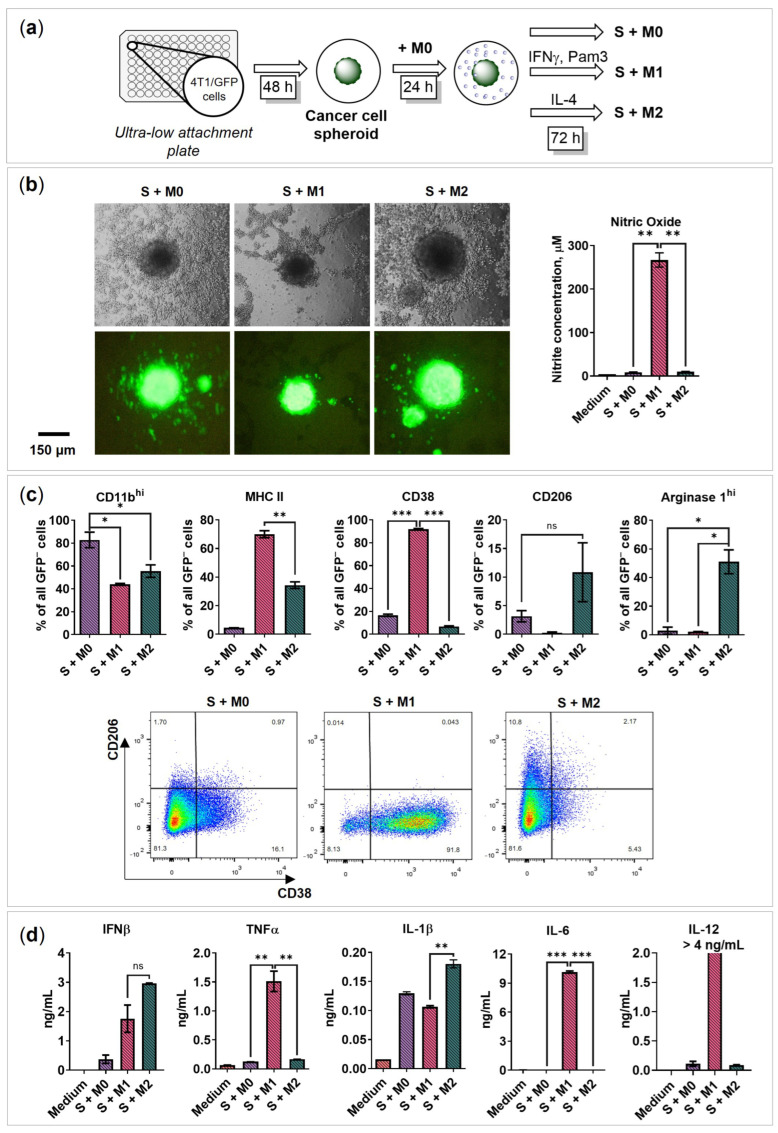
BMDMs programming in the presence of 4T1/GFP cancer cell spheroid under 3D conditions. (**a**) Experimental design for the generation of the 3D cancer cell spheroid model with free-floating M0, M1, and M2 macrophages. First, 4T1/GFP murine mammary cancer cells were plated in a 96-well black round bottom ultra-low attachment plate to achieve spheroids (S). After 48 h, BMDMs were added to the cancer cell spheroids. Cells were co-cultivated for 24 h to achieve TAM-like features, and then the factors were added to polarize cells towards M0, M1, and M2 phenotypes. After 72 h, co-cultures S + M0, S + M1, and S + M2 were analyzed. (**b**) Nitric oxide test, bright-field microscopy, and fluorescence microscopy of the 3D cancer cell model with M0, M1, and M2 macrophages 72 h after activation; cancer cells express GFP. (**c**) Flow cytometry determined marker expression of 3D polarized macrophages 72 h after activation; data are presented as % of GFP-negative (GFP^−^) cell population. Flow cytometry analysis showing representative staining of CD38 on the *x*-axis and CD206 on the *y*-axis. (**d**) The concentration of secreted cytokines in the media of S + M0, S + M1, and S + M2 72 h after activation. S—spheroid; medium—RPMI-1640 medium containing 10% FBS, 50 U/mL penicillin, 50 µg/mL streptomycin, 2 mM L-glutamine, and 10% L929-CM. Data are shown as the mean of two experiments ± SEM (*n* = 2, each experiment is a pool of 4 biological replicates). * *p* < 0.05; ** *p* < 0.01; *** *p* < 0.001; ns—non-significant.

**Figure 4 ijms-24-10763-f004:**
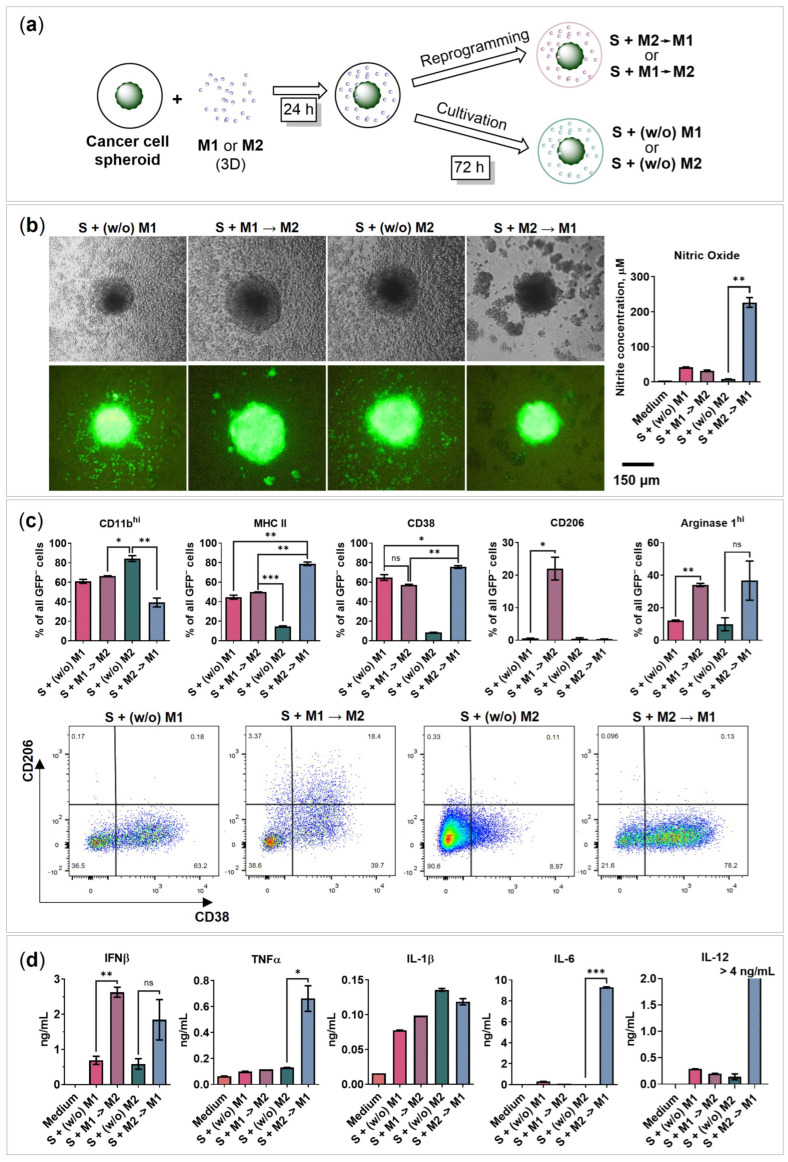
BMDMs reprogramming in the presence of 4T1/GFP cancer cell spheroid. (**a**) Experimental design for the reprogramming of free-floating M1 and M2 macrophages in the presence of cancer cell spheroid. BMDMs were activated for 48 h in 3D free-floating conditions to achieve M1 and M2 phenotypes. In parallel, 4T1/GFP murine mammary cancer cells were plated in a 96-well black round bottom ultra-low attachment plate to generate cancer cell spheroids (48 h). The activated M1 and M2 were added to the 4T1/GFP spheroids. Cells were co-cultivated for 24 h to achieve TAM-like features, and then the factors were added to reprogram cells towards M1→M2 (IL-4) and M2→M1 (IFNγ/Pam3SCK4) phenotypes. The M1 and M2 macrophages added to the 4T1/GFP spheroids without (*w*/*o*) reprogramming factors are indicated as S + (*w*/*o*) M1 and S + (*w*/*o*) M2, respectively. After 72 h, co-cultures S + (*w*/*o*) M1, S + M1→M2, S + (*w*/*o*) M2, and S + M2→M1 were analyzed. (**b**) Nitric oxide test, bright-field microscopy, and fluorescence microscopy of the 3D cancer cell spheroids and macrophages 72 h after reprogramming; cancer cells express green fluorescent protein gene (GFP). (**c**) Flow cytometry analysis of markers of reprogrammed macrophages in total GFP-negative (GFP^−^) cell population 72 h after reprogramming; data are presented as % of GFP^–^ cells. Flow cytometry analysis of CD38 on the *x*-axis and CD206 on the *y*-axis. (**d**) Analysis of cytokines in the media of S + (*w*/*o*) M1, S + M1→M2, S + (*w*/*o*) M2, and S + M2→M1 72 h after reprogramming. S—spheroid; medium—RPMI-1640 medium containing 10% FBS, 50 U/mL penicillin, 50 µg/mL streptomycin, 2 mM L-glutamine, and 10% L929-CM. Data are shown as the mean of two experiments ± SEM (*n* = 2, each experiment is a pool of 4 biological replicates). * *p* < 0.05; ** *p* < 0.01; *** *p* < 0.001; ns—non-significant.

**Figure 5 ijms-24-10763-f005:**
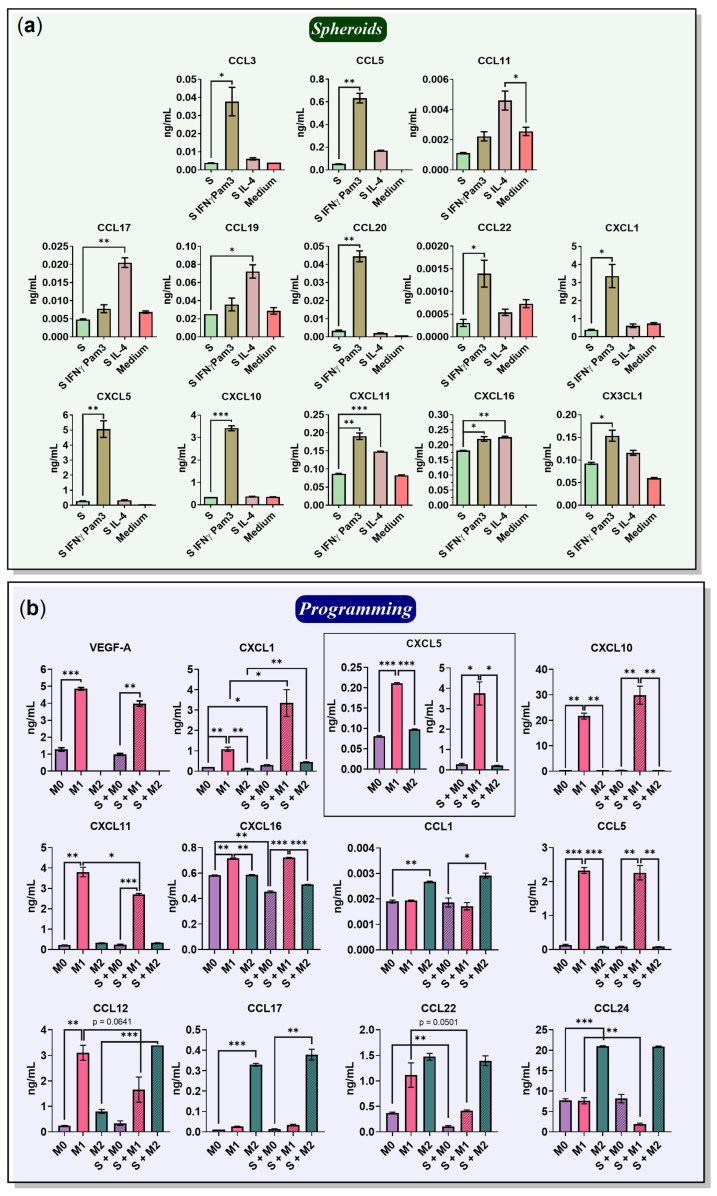
Chemokines produced by programmed macrophages and 4T1/GFP cancer cell spheroid under 3D free-floating conditions. Cell-conditioned medium from 3D monocultures and co-cultures was collected 72 h after programming. Then, the chemokines were determined using Luminex or ELISA. (**a**) Chemokines produced by 4T1/GFP cancer cell spheroid without stimulation and upon stimulation with IL-4 or IFNγ/Pam3SCK4. (**b**) Chemokines found in the medium of programmed macrophages in the presence (S + M) and absence (M) of cancer cell spheroids. S—spheroid; (*w*/*o*)—preprogrammed macrophages cultured without programming factors; medium—RPMI-1640 medium containing 10% FBS, 50 U/mL penicillin, 50 µg/mL streptomycin, 2 mM L-glutamine, and 10% L929-CM. Data are shown as the mean of two experiments ± SEM (*n* = 2, each experiment is a pool of 4 biological replicates). * *p* < 0.05; ** *p* < 0.01; *** *p* < 0.001.

**Figure 6 ijms-24-10763-f006:**
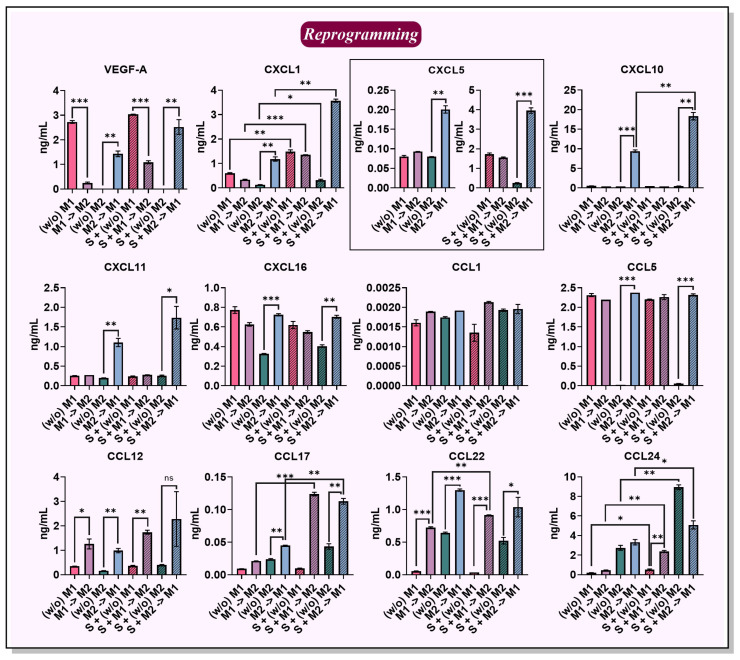
Chemokines in the medium of reprogrammed macrophages in the presence (S + M) and absence (M) of cancer cell spheroids. Cell media from 3D monocultures and co-cultures were collected 72 h after reprogramming. Then, the chemokines were determined using Luminex and ELISA. S—spheroid; (*w*/*o*)—preprogrammed macrophages cultured without programming factors. Data are shown as the mean of two experiments ± SEM (*n* = 2, each experiment is a pool of 4 biological replicates). * *p* < 0.05; ** *p* < 0.01; *** *p* < 0.001; ns—non-significant.

**Figure 7 ijms-24-10763-f007:**
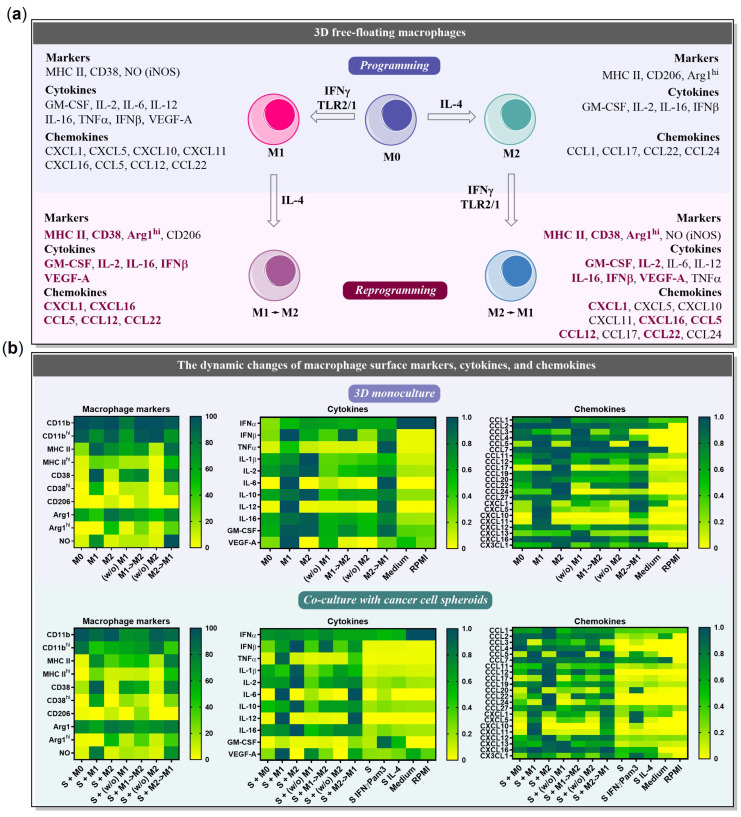
Summary of cell surface marker and secretome profiles of programmed and reprogrammed free-floating 3D macrophages. (**a**) Markers, cytokines, and chemokines expressed by both reprogrammed phenotypes M1→M2 and M2→M1 are highlighted in purple. iNOS—inducible NO synthase. (**b**) Heatmap of chemokines produced by BMDMs and 4T1/GFP cancer cell spheroids under 3D free-floating conditions. M0—undifferentiated BMDMs; M1—BMDMs programmed with IFNγ/Pam3SCK4; M2—BMDMs programmed with IL-4; (*w*/*o*) M1 and (*w*/*o*) M2—pre-programmed macrophages cultivated without (*w*/*o*) activation factors IFNγ/Pam3SCK4 and IL-4, respectively; M1→M2: M1 macrophages reprogrammed to M2 with IL-4 (IFNγ/Pam3SCK4 was removed); M2→M1: M2 macrophages reprogrammed to M1 with IFNγ/Pam3SCK4 (IL-4 was removed); S—4T1/GFP cancer cell spheroids; medium—RPMI-1640 medium supplemented with 10% FBS, 50 U/mL penicillin, 50 µg/mL streptomycin, 2 mM L-glutamine, and 10% L929-CM, representing the BMDMs cultivation medium; RPMI—RPMI medium without supplements. Scale: in the case of macrophage markers—the percent of maximal expression (0–100%); in the case of cytokines and chemokines—the normalized amount of secreted cytokine (0–1).

**Figure 8 ijms-24-10763-f008:**
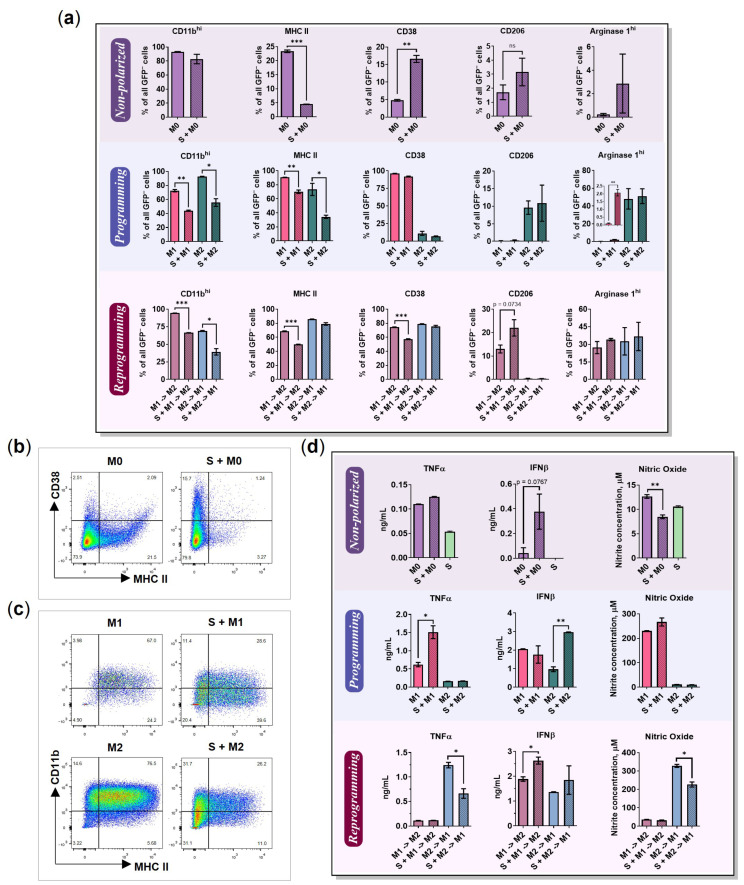
4T1 breast cancer cell spheroids affect macrophages. Macrophages were co-cultivated with breast cancer cell spheroids 24 h before programming or reprogramming and 72 h after programming or reprogramming. M1 phenotype was achieved by the addition of IFNγ/Pam3SCK4, M2 phenotype was achieved by the addition of IL-4. (**a**) Changes in cell marker levels of macrophages cultivated in the presence (S + M) and absence (M) of 4T1/GFP cancer cell spheroids. Cells have been analyzed using flow cytometry. (**b**) Flow cytometry analysis of MHC II on the *x*-axis and CD38 on the *y*-axis in GFP-negative cell population. (**c**) Flow cytometry analysis of MHC II on the *x*-axis and CD11b on the *y*-axis in GFP-negative cell population. (**d**) Changes in the levels of secreted cytokines of macrophages cultivated in the presence (S + M) and absence (M) of cancer cell spheroids detected by Luminex/ELISA. Data are shown as the mean of two experiments ± SEM (*n* = 2, each experiment is a pool of 4 biological replicates). * *p* < 0.05; ** *p* < 0.01; *** *p* < 0.001; ns—non-significant.

**Figure 9 ijms-24-10763-f009:**
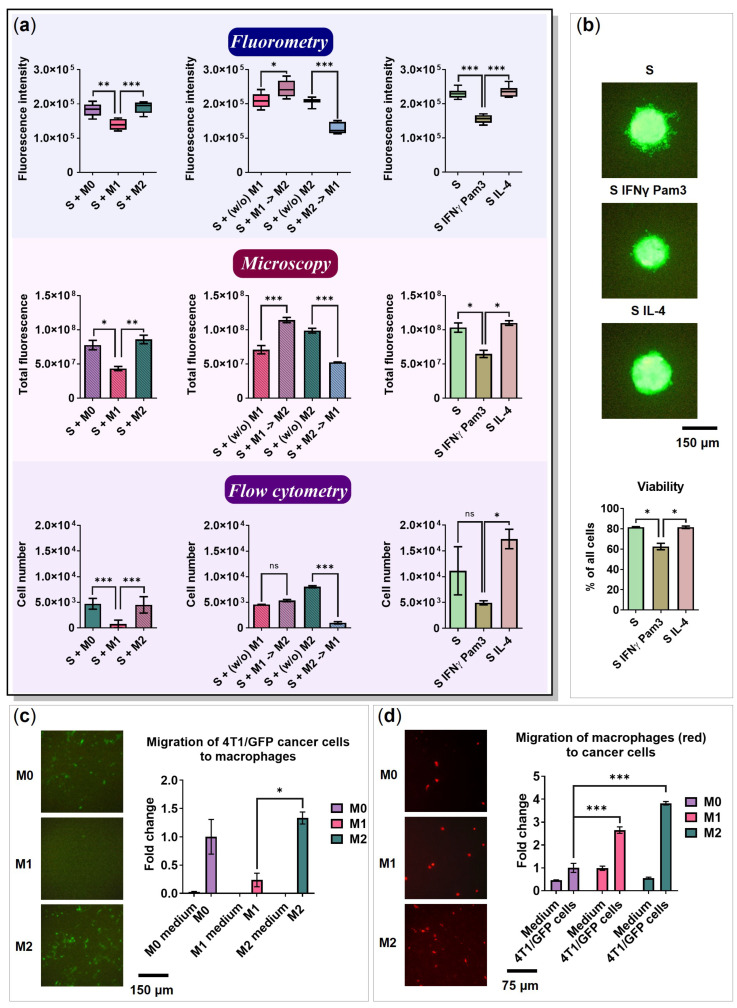
The size of the 4T1/GFP cell spheroids (**a**,**b**) and cell migration (**c**,**d**). (**a**) The size of the spheroids cultivated with different types of macrophages is determined using fluorometry (*n* = 6), microscopy (*n* = 3), and flow cytometry (*n* = 2, each experiment is a pool of 4 biological replicates). The 4T1/GFP cell spheroids were cocultured with macrophages, as shown in Figure 3a and Figure 4a, and analyzed on Day 6. (**b**) Representative fluorescent microscopy images and viability (flow cytometry) of cancer cell spheroids cultivated with IFNγ/Pam3SCK4 or IL-4. (**c**) Representative fluorescent microscopy images and fold change of 4T1/GFP cancer cells migrated towards M0, M1, and M2 macrophages through 8.0 μm pore inserts (*n* = 3). The cancer cell migration towards respective media (M0, M1, and M2) was used as a control. (**d**) Representative fluorescent microscopy images and fold change of M0, M1, and M2 macrophages migrated towards 4T1/GFP cancer cells through 8.0 μm pore inserts (*n* = 3). Macrophages have been labeled with CellTracker™ CM-DiI fluorescent dye (red). The macrophage migration to the 4T1/GFP cell culture medium was used as a control. Data are shown as mean ± SEM. * *p* < 0.05; ** *p* < 0.01; *** *p* < 0.001; ns—non-significant.

**Table 1 ijms-24-10763-t001:** Summary of changes in cell surface markers and secretome profile of macrophages cultivated in the presence of cancer cell spheroids compared to the same macrophages. Markers whose changes have been detected for four or more of the studied phenotypes (groups) are highlighted in bold. GM-CSF-induced chemokines are highlighted in blue. The decrease in 4T1 cell receptor CCR4, CXCR4, and CXCR7 ligands is highlighted in orange. (*w*/*o*)—cells incubated washed out activating factors (without IFNγ/Pam3SCK4, or IL-4), ND—changes not detected, **↑**—upregulated, **↓**—downregulated.

Group		Macrophage Markers	Cytokines	Chemokines
S + M0	upregulated	CD38 **↑**	**IFNβ ↑**	CXCL1 **↑**
	downregulated	**MHC II ↓**	NO (iNOS) **↓**	CCL22**↓**, CXCL12**↓**, CXCL16 **↓**
S + M1	upregulated	Arg1^hi^ **↑**	TNFα **↑**	CCL4 **↑**, CCL12 **↑**, CCL22 **↑**
	downregulated	**MHC II ↓**, **CD11b^hi^ ↓**	IL-1β **↓**	CCL24 **↓**, CXCL11 **↓**
S + M2	upregulated	*ND*	**IFNβ ↑**	CCL3 **↑**, CCL12 **↑**, **CXCL1 ↑**
	downregulated	**MHC II ↓**, **CD11b^hi^ ↓**	IL-12 **↓**	CXCL16 **↓**
S + (*w*/*o*) M1	upregulated	*ND*	*ND*	CCL20 **↑**, **CXCL1 ↑**
	downregulated	*ND*	*ND*	CCL3 **↓**, CXCL13 **↓**
S + M1→M2	upregulated	CD206 **↑**	**IFNβ ↑**	CCL17 **↑**, CCL20 **↑**, CCL22 **↑**, CCL24 **↑**, **CXCL1 ↑**
	downregulated	**MHC II ↓**, **CD11b^hi^ ↓**, CD38 **↓**	*ND*	*ND*
S + (*w*/*o*) M2	upregulated	*ND*	IL-1β **↑**, **IFNβ ↑**, TNFα **↑**	CCL24 **↑**, **CXCL1 ↑**
	downregulated	*ND*	*ND*	*ND*
S + M2→M1	upregulated	*ND*	*ND*	CCL17 **↑**, CCL24 **↑**
	downregulated	**CD11b^hi^ ↓**	NO (iNOS) **↓**, TNFα **↓**	*ND*

## Data Availability

The data presented in this study are available upon request from the corresponding authors.

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
