# Peer review of "Establishment and Characterization of Free-Floating 3D Macrophage Programming Model in the Presence of Cancer Cell Spheroids"

_ijms, 2023, doi:10.3390/ijms241310763_

Round 1
Reviewer 1 Report
In this manuscript, Korotkaja et al. investigate the 3D model of cancer cell spheroid/macrophage co-culture under free-floating conditions can be used for studies of macrophage plasticity and for the development of targeted cancer immunotherapy.
It's an exciting study, but a few things need to be addressed to improve the paper:
1. It is recommended to seek professional help for data and statistical analysis.
2. Figure 10 and its description should be moved to the results section.
3. Figure 3 should be moved to the last part of the result section.
4. A large part of Figure 3, Table 1, and Figure 10 shows overlapping results. Figure 3, Table 1, and Figure 10 should be combined into one figure.
5. Significance must be marked on all Figures.
6. In Figure 1, the reason or meaning of IL-1beta expression increased more in M2 than in M1 should be discussed.
Reviewer 2 Report
The manuscript entitled "Establishment and Characterization of Free-Floating 3D Macrophage Programming Model in the Presence of Cancer Cell Spheroids" attempts to develop and characterize a new in vitro 3D model for macrophage programming in the presence of cancer cell spheroids. The manuscript is well-written and organized in proper sections. Some minor questions and comments are stated below and should be addressed before considering the publication of this manuscript in International Journal of Molecular Sciences:
1. Please consider introducing a list of abbreviations in the beginning of the manuscript.
2. Consider highlighting the novelty of the study to the research field (lines 64-72).
3. The statistical analysis seems to be lacking in some figures. Please provide it for all figures.
4. In the subsection 4.2, what were the cell passages used? Please indicate it.
5. In conclusion, please consider discussing future perspectives in this research field, as well as advantages and disadvantages of the present study.
6. Revise all references formatting words such as "in vitro" and "in vivo" in italic letter.
Minor comments:
- Line 20: Add "and" between M1 and M2 and delete the comma "(M1 and M2)".
- Line 94: Format "p" from probability values in italic letter. Revise it along the manuscript.
- Line 117: Format "n" (for example, "n =3") in italic letter. Revise it along the manuscript.
- Line 123: Define "SEM" where it first appears in the manuscript.
- Lines 529-533: Please format these sentences in non-italic letter.
- Line 754: Define "RPMI" where it first appears in the manuscript.
- Line 762: Format "®" from "ATCC®" above the line.
- Line 780: Define "ACK" where it first appears in the manuscript.
- Line 792: Revise the format of "3.5∙104" replacing by "3.5x104". Revise all the manuscript for similar numbers.
- Line 835: Define the abbreviation "RT" where it first appears in the manuscript.
- Lines 881-882: Use the abbreviation "NO" instead of "nitric oxide" since this abbreviation is already previously defined.
Round 2
Reviewer 1 Report
None